# Interpreting the Synchronization Gap: The Hidden Mechanism Inside Diffusion Transformers

## Abstract

Recent theoretical models of diffusion processes, conceptualized as coupled Ornstein-Uhlenbeck systems, predict a hierarchy of interaction timescales, and consequently, the existence of a synchronization gap between modes that commit at different stages of the reverse process. However, because these predictions rely on continuous time and analytically tractable score functions, it remains unclear how this phenomenology manifests in the deep, discrete architectures deployed in practice. In this work, we investigate how the synchronization gap is mechanistically realized within pretrained Diffusion Transformers (DiTs). We construct an explicit architectural realization of replica coupling by embedding two generative trajectories into a joint token sequence, modulated by a symmetric cross attention gate with variable coupling strength $g$. Through a linearized analysis of the attention difference, we show that the replica interaction decomposes mechanistically. We empirically validate our theoretical framework on a pretrained DiT-XL/2 model by tracking commitment and per layer internal mode energies. We validate our findings on different ImageNet classes, image sizes (256 and 512) as well as different models PixArt-$\Sigma$. Our results reveal that: (1) the synchronization gap is an intrinsic architectural property of DiTs that persists even when external coupling is turned off, not an artifact of the imposed coupling; (2) as predicted by our spatial routing analysis, the gap completely collapses under strong coupling $g \to 1$; (3) the gap is strictly depth localized, emerging sharply only within the final layers of the Transformer; and (4) global, low frequency structures consistently commit before local, high frequency details. Ultimately, our findings provide a mechanistic account of the difference dynamics in Diffusion Transformers, isolating speciation transitions to the terminal layers of the network.

## 1 Introduction

Diffusion models generate data by learning to reverse a stochastic noising process, progressively transforming Gaussian noise into structured samples (Sohl-Dickstein, Weiss, Maheswaranathan, and Ganguli, 2015; Ho, Jain, and Abbeel, 2020; Song, Sohl-Dickstein, Kingma, Kumar, Ermon, and Poole, 2020b; Lai, Song, Kim, Mitsufuji, and Ermon, 2025). Among current architectures, Diffusion Transformers (DiTs) (Peebles and Xie, 2023) have emerged as the foundational standard for generative modeling. By replacing the rigidly structured convolutional U-Net (Ronneberger, Fischer, and Brox, 2015) with a highly scalable sequence of Transformer blocks (Vaswani, Shazeer, Parmar, Uszkoreit, Jones, Gomez, Kaiser, and Polosukhin, 2017) that operate on modality agnostic patchified latent tokens (Rombach, Blattmann, Lorenz, Esser, and Ommer, 2022), DiTs have rapidly expanded beyond their origins in high fidelity image synthesis. This architecture now powers SOTA multimodal systems across highly specialized domains ranging from healthcare applications (Kazerouni, Aghdam, Heidari, Azad, Fayyaz, Hacihaliloglu, and Merhof, 2023), spatiotemporal video generation (Ho, Chan, Saharia, Whang, Gao, Gritsenko, Kingma, Poole, Norouzi, Fleet, and Salimans, 2022; Brooks, Peebles, Holmes, DePue, Guo, Jing, Schnurr, Taylor, Luhman, Luhman, Ng, Wang, and Ramesh, 2024), and 3D molecular drug design (Alakhdar, Póczos, and Washburn, 2024) to complex statistical applications in causal inference (Sanchez and Tsaftaris, 2022; Ma, Melnychuk, Schweisthal, and Feuerriegel, 2024). However, despite this unprecedented empirical success, the internal mechanisms by which these models resolve gen-

erative ambiguity when transitioning from unstructured noise to specific, coherent representations remains poorly understood from an interpretability perspective.

The necessity for interpretability (Linardatos, Papastefanopoulos, and Kotsiantis, 2020) in deep learning stems from a fundamental trade off between predictive capacity and human comprehension. In basic statistical models like linear regression, the mapping is intrinsically interpretable: every learned weight directly quantifies the influence of a feature on the output, allowing practitioners to easily audit decisions and extract true causal relationships. However, as architectures scale into deep, nonlinear neural networks, this transparency degrades into a black box. Overcoming this opacity is not merely a post hoc diagnostic exercise, but a critical engineering requirement where interpreting internal representations allows developers to debug failure modes, predict edge case behaviors, and design more efficient, reliable architectures. Furthermore, interpretability is a strict prerequisite for deployment in high stakes domains such as healthcare, where algorithmic accountability, bias mitigation, and safety are legally and ethically mandated (Goktas and Grzybowski, 2025; Ennab and Mcheick, 2024). This need is particularly acute in the natural sciences. As physicists, biologists, and chemists increasingly adopt foundational deep learning models, they require more than just a highly accurate predictive black box, they require mechanistic insight.

A recent line of work based on nonequilibrium statistical physics (Raya and Ambrogioni, 2023; Biroli, Bonnaire, Bortoli, and Mézard, 2024; Kamb and Ganguli, 2024; Sclocchi, Favero, and Wyart, 2025) has begun to address these interpretability questions by identifying sharp dynamical transitions in the reverse generative process. In Biroli et al. (2024), the authors established that diffusion models trained on structured data distributions undergo two macroscopic phase transitions, a speciation time, at which the trajectory commits to a particular data mode, and a collapse time, at which it locks onto a specific training example. These transitions have since been characterized through replica analysis of Gaussian mixtures (Biroli, Bonnaire, Bortoli, and Mézard, 2024), extended to general class structures via free entropy criteria (Achilli, Benedetti, Biroli, and Mézard, 2026), and connected to entropic signatures observable in trained models (Handke, Stančević, Koulischer, Demeester, and Ambrogioni, 2026). In the setting of coupled multimodal generation, Albrychiewicz et al. (2026) showed that modeling the interaction between two coupled diffusion trajectories as a pair of Ornstein–Uhlenbeck (OU) processes (Uhlenbeck and Ornstein, 1930) reveals a synchronization gap. In the case of symmetric coupling, this is a temporal window during which the common eigenmode has speciated while the difference eigenmode has not. This gap arises from a spectral hierarchy in the interaction timescales and has been shown to depend on the coupling strength $g$. The coupled OU processes approach has also been recently extended by Lu & Tang (2026) for breaking detailed balance which can be used to accelerate generative process.

These theoretical results, however, are formulated entirely in the language of continuous stochastic processes with analytically tractable score functions. In contrast, a pretrained Diffusion Transformer is not analytically tractable. It is a deep, discrete residual network in which the score function is implicitly defined by the composition of attention layers, pointwise nonlinearities, and adaptive normalization modules. The central question motivating this work is how the synchronization gap phenomenology is realized in the architecture of a Diffusion Transformer, and what mechanism is responsible for its existence?

We answer this question both theoretically and empirically. On the theoretical side, we construct an explicit mapping from the coupled OU system into the self-attention mechanism of a pretrained DiT. By embedding two generation trajectories into a single token sequence and introducing a blockwise normalized, symmetric attention gate which depends on the coupling strength $g$, we obtain a controlled architectural realization of replica coupling Equation 19. Linearizing the resulting difference in attention output around the symmetric state, where two replicas are equivalent, we decompose it into two mechanistically distinct terms Equation 25. The first term, which we refer to as a spatial routing, is the one where unperturbed attention kernel transports a perturbed value signal across token positions, and in the second, the perturbation enters through the softmax Jacobian of the attention weights themselves. We show that these two channels are suppressed by different functions of the coupling strength, $\frac{1-g}{1+g}$ and $\frac{1}{1+g}$, respectively, and argue that the first term is dominant for low frequency modes of replicas difference.

To move beyond the linear regime and to determine a speciation time, we model the local distribution of the replicas difference modes as a symmetric two component Gaussian mixture, following the approach of Biroli

et al. (2024); Albrychiewicz et al. (2026) adapted to the discrete block structure. Projecting the resulting fixed point Equation 41 onto empirical eigenmodes of the initial difference covariance yields a scalar self consistency condition for each mode Equation 49, with a modewise speciation parameter that decomposes into an attention gated signal-to-noise ratio (SNR) Equation 53. We also briefly mention how Renormalization Group flow (Wilson, 1975) approach can be used to analyze the Transformer architecture. In statistical physics, the Renormalization Group is a mathematical apparatus used to study phase transitions by averaging out microscopic degrees of freedom to find macroscopic laws. The synchronization gap Equation 66, the difference in speciation times between leading and trailing modes is then shown to scale as $\mathcal{O}(\frac{1-g}{1+g})$ under an assumption that spatial routing term dominates, predicting its shrinkage at strong coupling.

On the empirical side, we test these predictions on a pretrained DiT-XL/2 model (Peebles and Xie, 2023) using two complementary experimental protocols. The first protocol section 3.1 measures when the model behaviorally commits by coupling two replicas for an initial steps and then letting them evolve independently, measuring the agreement of the final decoded images through feature space cosine similarity (using pretrained ResNet-50 encoder (He, Zhang, Ren, and Sun, 2016)) and scale dependent pixel discrepancies. The second protocol section 3.2, with a sweep across all 28 Transformer layers, identifies where this commitment is represented internally by tracking the empirical mode energies of the hidden state replica difference, evaluated at the speciation time identified by Protocol I. To confirm our findings are not an artifact of a particular model and image class, we conduct experiments on an additional ImageNet class (Deng, Dong, Socher, Li, Li, and Fei-Fei, 2009) and on the other model PixArt-$\Sigma$-256 (Chen, Ge, Xie, Wu, Yao, Ren, Wang, Luo, Lu, and Li, 2024).

Our main empirical findings are:

1. The synchronization gap exists in DiT even without coupling. At $g = 0$, the Protocol II experiment reveals a clear separation between leading and trailing modes energies concentrated in the final layers of Transformer cf., figure 1. This demonstrates that the gap is an intrinsic property of pretrained DiT architecture, not just an artifact of the imposed coupling.

2. The gap collapses in the strong coupling region as predicted in theory section cf., figure 5a–figure 5c. As $g$ increases from 0 to 1, the internal leading and trailing mode separation is progressively suppressed. This is consistent with theoretical prediction that, under the assumption of spatial routing term dominance which we confirm empirically, the spectral hierarchy collapses as $g \to 1$.

3. The gap is depth localized. In the weakly coupled regime, the synchronization gap is near zero in early layers and exhibits a transient texture inversion in middle layers, where trailing modes temporarily stabilize before leading ones. The gap emerges sharply only within the last $\approx 5$ layers. This identifies the terminal layers as the site where the network performs frequency based routing.

4. Global structure commits before local detail. The scale dependent probe of Protocol I confirms that low frequency image structure stabilizes substantially earlier than high frequency details across all tested coupling strengths $g$ cf., figure 4a–figure 4f.

The remainder of this paper is organized as follows. In section 2 we develop the theoretical framework, from the OU motivation through the attention gated propagator to the modewise SNR formula and gap collapse prediction. We follow with section 3 in which we describe the two empirical protocols and their implementation. In section 4 we present the experimental results. We conclude with section 5 where we discuss the findings and their implications on recent training free acceleration methods and other applications. In Appendix A we provide the detailed derivation of the attention difference between replicas, and in Appendix B we establish the spatial routing term dominance for low frequency modes by a combination of exact identities and direct measurement.

## 2  Theoretical Framework

We develop an effective theoretical framework connecting the continuous statistical physics of coupled diffusion processes to the discrete architecture of Diffusion Transformers. The goal is not to claim an exact

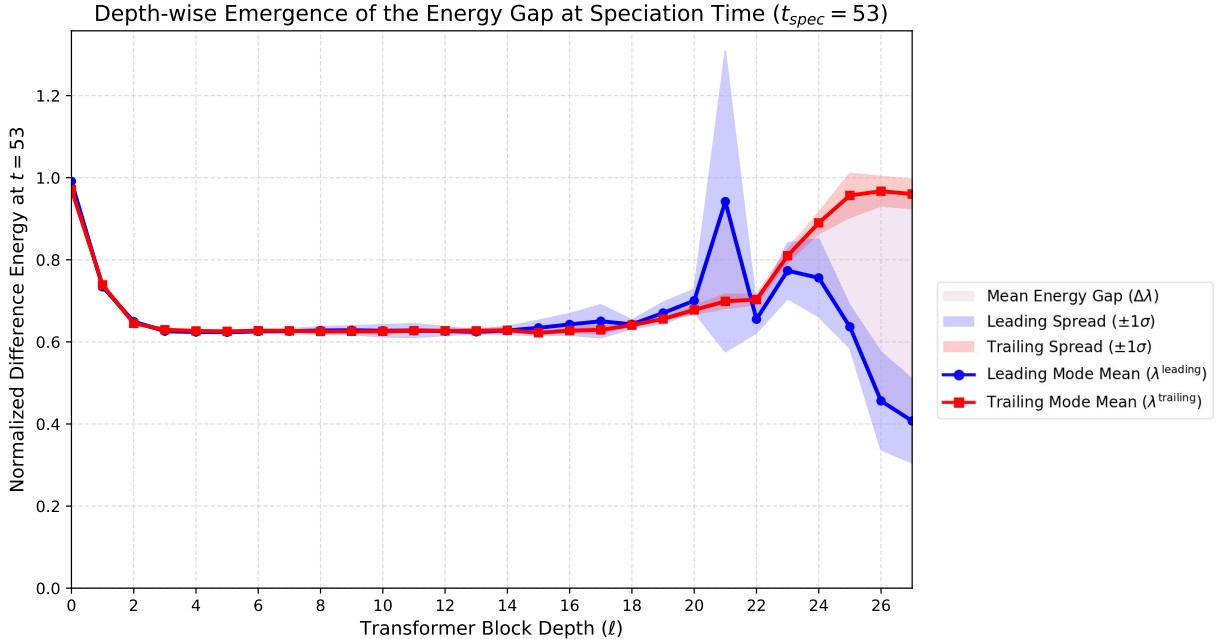

Figure 1: For coupling strength $g = 0$ and speciation step $s = 53$, we sweep across all Transformer layers to evaluate the normalized fixed basis energies of leading and trailing internal difference modes Equation 78–Equation 79. Even with coupling turned off, the gap is present at deep layers of DiT.

equivalence between the OU description and a pretrained transformer, but rather to extract a testable mechanistic hypothesis. The synchronization gap phenomenology predicted in coupled diffusion theory is implemented in DiTs through spatially selective routing in self-attention, and can be revealed by controlled replica coupling. In contrast to the case discussed in Albrychiewicz et al. (2026), the synchronization gap that we consider is a temporal window between the commitment times of distinct projected modes of the replica difference channel. In the empirical analysis, these modes are defined as fixed principal directions of the initial empirical difference covariance, and the measured observables are their time dependent projected energies.

## 2.1 Coupled Reverse Diffusion: Motivation and Limitations of the Linear Theory

Unless stated otherwise, we will consider the diffusion process as taking values in the latent space of dimension $d_z$. The variance preserving (VP) forward diffusion process is an OU process which has an Ito SDE realization as

$$dz_t = -\frac{1}{2}\beta_t z_t dt + \sqrt{\beta_t} dW_t,\tag{1}$$

where $z_0 \sim p_{\text{data}}$, $\beta_t$ is the noise schedule and $W_t$ a standard Wiener process in $R^{d_z}$. From a physics perspective, this SDE describes the overdamped Langevin dynamics of a particle coupled to a heat reservoir. Under this process, the conditional forward marginal given $z_0$ is

$$q(z_t \mid z_0) \sim \mathcal{N}(\alpha_t z_0, \sigma_t^2 I),\tag{2}$$

where

$$\alpha_t = e^{-\frac{1}{2}\int_0^t \beta_s ds}, \quad \text{and} \quad \sigma_t^2 = 1 - \alpha_t^2.\tag{3}$$

Following Albrychiewicz et al. (2026), we study two replicas $z_t^A, z_t^B \in R^{d_z}$ which are two realizations/trajectories of an identical stochastic process. Here, the superscript $A, B$ are distinct labels, not a

free index. These replicas are constructed in such a way that their reverse time dynamics are coupled with a symmetric relaxation matrix with a strength $g \geq 0$. In the continuous OU approximation, the coupled reverse process is

$$dz_t^A = \left[ f(z_t^A, t) + g(z_t^A - z_t^B) \right] dt + \sqrt{\beta_t} d\bar{W}_t^A, \tag{4}$$

$$dz_t^B = \left[ f(z_t^B, t) + g(z_t^B - z_t^A) \right] dt + \sqrt{\beta_t} d\bar{W}_t^B, \tag{5}$$

where the reverse drift is

$$f(z_t, t) = -\frac{1}{2}\beta_t z_t - \beta_t \nabla_{z_t} \log p_t(z_t) \tag{6}$$

and $s(z_t, t) = \nabla_{z_t} \log p_t(z_t)$ is the score. The reverse process is initialized at the terminal time $t = T$ from the stationary Gaussian distribution $z_T^A, z_T^B \sim \mathcal{N}(0, I)$ with the convention that the time orientation flows backwards $dt < 0$. We assume that the reverse Wiener processes $\bar{W}_t^A$ and $\bar{W}_t^B$ are independent white noise.

Similarly to the approach used in Albrychiewicz et al. (2026), we introduce common and difference modes

$$u_t = \frac{z_t^A + z_t^B}{\sqrt{2}}, \qquad v_t = \frac{z_t^A - z_t^B}{\sqrt{2}}. \tag{7}$$

However, in this case, we cannot use the exact score and these modes do not fully decouple the system given by Equation 4-Equation 5. Nevertheless, at the beginning of the reverse process, when noise level is high, we can approximate the marginal. If one approximates the marginal score by a single Gaussian $\mathcal{N}(0, \Sigma_t)$ then the score is linear

$$\nabla_z \log p_t(z) \approx -\Sigma_t^{-1} z, \tag{8}$$

in that regime, the dynamics decouple

$$\mathrm{d}u_t = \left[ -\tfrac{1}{2}\beta_t u_t + \beta_t \Sigma_t^{-1} u_t \right] \mathrm{d}t + \sqrt{\tfrac{\beta_t}{2}} \, \mathrm{d}(\bar{W}_t^A + \bar{W}_t^B), \tag{9}$$

$$\mathrm{d}v_t = \left[ -\tfrac{1}{2}\beta_t v_t + \beta_t \Sigma_t^{-1} v_t + 2g \, v_t \right] \mathrm{d}t + \sqrt{\tfrac{\beta_t}{2}} \, \mathrm{d}(\bar{W}_t^A - \bar{W}_t^B). \tag{10}$$

The common mode $u_t$ evolves identically to a single uncoupled trajectory. The difference mode $v_t$ acquires an additional restoring term $2g \, v_t$ that accelerates its drift toward zero mean hence acting as a restoring force that pulls the replicas together.

The linear theory predicts a hierarchy of damping rates across directions in latent space. However, by itself it does not identify those directions with spatial frequencies, nor can it generate branch formation. As long as the linearized operator remains stable, the unique fixed point for the difference mode is $v = 0$ [1]. Hence the linear Gaussian theory is best understood as a motivation for a difference channel hierarchy, not yet as a complete account of speciation.

The difference channel hierarchy is the origin, in the linearized approximation, of the synchronization gap of Albrychiewicz et al. (2026). In the coupled-OU model the difference mode $v$ carries the extra restoring rate $2g$ (Equation 10), its eigenvectors relax on a hierarchy of timescales, and the slowest commits last. Our task is to realize this hierarchy inside a pretrained DiT, whose score is not analytic, and to measure it (section 2.6).

This motivates two extensions. First, we want to implement an architectural realization of symmetric coupling inside DiT self-attention. Unlike a simple linear penalty $g(z^A - z^B)$, joint self-attention inherently introduces non-linearity through its softmax activation and multiplicative query-key interactions. By allowing the tokens of replica $A$ to attend to the tokens of replica $B$, the effective coupling strength becomes state dependent which brings us to the realm of nonlinear SDEs or SDEs with multiplicative noise. Second, from a distributional perspective, we want to consider the simplest non-linear marginal probability that breaks the linear score assumption and allows stable branch formation in the difference channel. To generate branch formation, the system must undergo a symmetry breaking bifurcation which would result in a nontrivial fixed point $v \neq 0$. This mathematically requires the drift to exhibit a localized instability at the origin, which can only occur if the score function $\nabla_z \log p(z)$ contains higher order nonlinearities.

---

[1]For later discussion we suppress index $t$ for clarity.

## 2.2 Discrete DiT Model Architecture

To parameterize the reverse dynamics, we use the DiT architecture (Peebles and Xie, 2023). A DiT operates on a sequence of latent tokens rather than on a spatial feature pyramid as in a convolutional U-Net. We begin with a noisy spatial latent

$$z_t \in \mathbb{R}^{C \times H \times W}, \tag{11}$$

where $C$, $H$, and $W$ denote the latent channel, height, and width dimensions. We partition $z_t$ into $N = (H/p)(W/p)$ non overlapping patches of size $C \times p \times p$, assuming $p \mid H$ and $p \mid W$, we flatten each patch into a vector in $\mathbb{R}^{Cp^2}$, and map it through a learned patch embedding $W \in \mathbb{R}^{d_{\mathrm{model}} \times Cp^2}$ to a token in $\mathbb{R}^{d_{\mathrm{model}}}$. This yields a token sequence of length $N$

$$X_t \in \mathbb{R}^{N \times d_{\mathrm{model}}}, \qquad N = (H/p)(W/p). \tag{12}$$

The Transformer maps this sequence to an output sequence of the same length, conditioned on the diffusion timestep $t$ and external conditioning $c$ through adaptive layer normalization. A final linear decoder followed by an unpatchifying reshape maps the output tokens back to $\mathbb{R}^{C \times H \times W}$ to produce the noise prediction $\hat{\epsilon}_\theta(z_t, t, c)$. In the DiT-XL/2 configuration, $p = 2$ and $d_{\mathrm{model}} = 1152$.

Let $H_0 = X_t$. Each Transformer block then takes the form

$$\tilde{H}_\ell = H_\ell + \alpha_\ell \odot \mathrm{Attn}_\ell\big(\mathrm{adaLN}_{1,\ell}(H_\ell, t, c)\big), \tag{13}$$

$$H_{\ell+1} = \tilde{H}_\ell + \beta_\ell \odot MLP_\ell\big(\mathrm{adaLN}_{2,\ell}(\tilde{H}_\ell, t, c)\big), \tag{14}$$

where $\mathrm{Attn}_\ell$ denotes multi-head self-attention, which mixes information across token positions, and $MLP_\ell$ is a tokenwise feed-forward network acting independently on each token. The adaptive layer-normalization modules $\mathrm{adaLN}_{1,\ell}$ and $\mathrm{adaLN}_{2,\ell}$ shift and scale normalized activations using embeddings derived from the diffusion timestep $t$ and conditioning $c$. The gating vectors $\alpha_\ell, \beta_\ell \in \mathbb{R}^{d_{\mathrm{model}}}$ are likewise predicted from the conditioning embeddings, broadcast across token positions, and initialized at zero to stabilize training.

For a replica pair $(A, B)$, we concatenate the two token sequences along the token dimension,

$$X_t = [X_t^A; X_t^B] \in \mathbb{R}^{2N \times d_{\mathrm{model}}}. \tag{15}$$

For each attention head, let $Q$, $K$, and $V$ denote the corresponding query, key, and value matrices, and let $d_h$ be the per-head hidden dimension. The pre-softmax attention logits are

$$S = \frac{QK^\intercal}{\sqrt{d_h}}. \tag{16}$$

Under the replica concatenation, these decompose into block form as

$$S = \begin{bmatrix} S_{AA} & S_{AB} \\ S_{BA} & S_{BB} \end{bmatrix}, \qquad V = \begin{bmatrix} V_A \\ V_B \end{bmatrix}. \tag{17}$$

Here $S_{AA}$ and $S_{BB}$ are the intra-replica attention logits, while $S_{AB}$ and $S_{BA}$ are the inter-replica logits. This decomposition will allow us to introduce attention gating by modulating the inter-replica interactions separately from the intra-replica ones.

A naive softmax over all $2N$ tokens would implicitly couple replicas through the partition function even when the coupling strength $g = 0$, introducing a confound factor. To ensure a clean baseline, we compute attention weights with separate row wise softmax normalizations within each block

$$[A_{ij}]_{mn} = \frac{\exp([S_{ij}]_{mn})}{\sum_{k=1}^N \exp([S_{ij}]_{mk})}, \tag{18}$$

where $m$ denotes query token and $n$ the key token within that block so $m, n \in \{1, 2, \cdots, N\}$ and $i, j$ label $A, B$. These attention weights can be organized in terms of block matrices corresponding to intra- and inter-replica interactions, as in Equation 17.

We realize the continuous coupling strength $g$ through a normalized mixture of intra replica and inter replica attention outputs

$$\text{Attn}_g(X) = \frac{1}{1+g} \left( \underbrace{\begin{bmatrix} A_{AA} V_A \\ A_{BB} V_B \end{bmatrix}}_{\text{intra}} + g \underbrace{\begin{bmatrix} A_{AB} V_B \\ A_{BA} V_A \end{bmatrix}}_{\text{inter}} \right). \tag{19}$$

This construction satisfies three essential properties: (i) at $g = 0$, the replicas are exactly decoupled; (ii) the coupling is perfectly symmetric under $A \leftrightarrow B$ exchange; and (iii) the weights $1/(1+g)$ and $g/(1+g)$ sum to 1, the mixture is normalized, so the residual stream scale does not grow trivially with $g$. This ensures that the pretrained network operates safely within its learned dynamic range. For stability, we keep $g$ in range $[0, 1]$. Equation 19 is our proposed architectural realization of symmetric coupling in a pretrained DiT. In this realization, the difference mode $v$ interaction is mediated by self-attention routing across the two replicas. Mathematically, however, it will appear as a perturbation to a special state that we will define below.

We emphasize that Equations 13–14, the transformer block whose response we analyze, are the standard, unmodified DiT, and all of $A_{ij}$, $V$ the MLP, and AdaLN are evaluated from the pretrained DiT-XL/2 weights. The only departure is the coupling layer Equations 18-19, which re-routes these standard outputs across the two replicas, yet it does not alter the block. In particular, the blockwise normalization Equation 18 leaves every single replica computation identical to ordinary self-attention and modifies only the inter-replica term, whose strength is the controlled parameter $g$.

The construction is the controlled probe at the center of our approach. A pretrained DiT is not analytically tractable, so the synchronization gap predicted by the continuous coupled-OU theory cannot be read off its score directly. Instead, we make it measurable by reintroducing the coupling as an architectural knob. Two trajectories share an initial noise $z_t$ and a tunable amount of mutual attention $g$, which lets us inject a symmetry breaking difference between them and track how the network propagates, amplifies, or suppresses it across depth and reverse time. The construction is built so that the limits are interpretable. At $g = 0$ the replicas are exactly decoupled Equation 19, so the pair is just two independent denoising runs sharing $z_t$ and any structure we observe is an intrinsic property of the pretrained DiT, not an artifact of the probe. As $g \to 1$ the symmetric mixture forces the two runs into agreement. The normalization $1/(1+g)$ keeps the residual stream scale fixed across this range, so a $g$-sweep isolates the effect of coupling rather than of changing activation magnitudes. Operationally, the gap measures whether the network fixes different spatial scales of a sample at different times, a depth- and time-resolved statement about how generative ambiguity is resolved, which is the question Sec. 4 answers.

## 2.3 Linearized Analysis of the Attention Difference

In this subsection we obtain the first order attention difference Equation 25, separating the response into the unperturbed attention kernel and a perturbation of the attention kernel, this separation is central for the rest of the theory and measurements.

Given the above DiT architecture, consider two identical per layer outputs of the attention mechanism $H_{\ell,0}^A = H_{\ell,0}^B$. We will refer to these identical outputs as the symmetric state. We want to study the per layer response to a symmetry breaking perturbation $h_\ell$ of this symmetric state. We then consider the deformed output of the attention mechanism

$$H_\ell^A = H_{\ell,0} + \frac{h_\ell}{\sqrt{2}}, \qquad H_\ell^B = H_{\ell,0} - \frac{h_\ell}{\sqrt{2}}. \tag{20}$$

Expanding the value projection to first order around the symmetric state, we get

$$V_\ell^A = V_{\ell,0} + \delta h_\ell + \mathcal{O}(\|h_\ell\|^2), \tag{21}$$

$$V_\ell^B = V_{\ell,0} - \delta h_\ell + \mathcal{O}(\|h_\ell\|^2), \tag{22}$$

with $\delta h_\ell = J_{\ell,V} h_\ell$ for an effective Jacobian operator $J_V$ that absorbs the per layer normalization and value projection.

Similarly, the key and query dot products induce perturbations in the attention weight matrices that respect the replica exchange symmetry $A \leftrightarrow B$ (which maps $h_\ell \to -h_\ell$)

$$A_{\ell,AA} = A_{\ell,0} + \delta A_\ell^{(+)} + \mathcal{O}(\|h_\ell\|^2), \qquad A_{\ell,BB} = A_{\ell,0} - \delta A_\ell^{(+)} + \mathcal{O}(\|h_\ell\|^2), \tag{23}$$

$$A_{\ell,AB} = A_{\ell,0} + \delta A_\ell^{(-)} + \mathcal{O}(\|h_\ell\|^2), \qquad A_{\ell,BA} = A_{\ell,0} - \delta A_\ell^{(-)} + \mathcal{O}(\|h_\ell\|^2), \tag{24}$$

where $\delta A_\ell^{(+)}$ and $\delta A_\ell^{(-)}$ are linear in $h_\ell$. They encode the Jacobians of the blockwise softmax acting on the perturbed intra replica and inter replica logits, respectively.

Substituting in Equation 19, to first order in $\|h_\ell\|$, the contribution of the coupled attention layer to the perturbation is

$$\mathrm{Attn}_{g,\ell}^A - \mathrm{Attn}_{g,\ell}^B = \frac{1-g}{1+g}\, 2\, A_{\ell,0}\, \delta h_\ell \;+\; \frac{1}{1+g}\, 2\big(\delta A_\ell^{(+)} + g\, \delta A_\ell^{(-)}\big) V_{\ell,0} \;+\; \mathcal{O}(\|h_\ell\|^2), \tag{25}$$

see Appendix A for details. This attention difference decomposes the linear response into two mechanistically distinct first order pathways i.e. one through the value map and one through the attention kernel.

The first term, $A_{\ell,0}\, \delta h_\ell$, is a value perturbation routed by the unperturbed attention kernel. The symmetric attention pattern $A_{\ell,0}$ is held fixed, while the perturbation perturbs the value vectors, and the resulting signal is transported across token positions. Consequently, we will occasionally call this the spatial routing term. In other words, this is the contribution from the value path with the attention map frozen. Its prefactor $(1-g)/(1+g)$ decreases with $g$ and vanishes at $g = 1$, so strong inter replica coupling suppresses this channel completely.

The second term, $\big(\delta A_\ell^{(+)} + g\, \delta A_\ell^{(-)}\big) V_{\ell,0}$, is a perturbation of the attention kernel. Here the perturbation enters through the query key softmax pathway, changing the attention weights themselves, which are then applied to the background values $V_{\ell,0}$. This is the Jacobian contribution from the attention pattern modulation channel. Unlike the routing term, it is suppressed only by $1/(1+g)$ and therefore remains nonzero even at $g = 1$. Thus increasing $g$ shifts the leading linear response from a fixed kernel routing of a perturbed value signal towards a perturbation of the attention kernel itself.

## 2.4 The Linearized Propagator

In this subsection, we package the per block effect of attention and the MLP on the difference mode into a single linear map across one Transformer block, so that its modal gains can later be tracked through depth. This map is the propagator $K_g$ which we derive.

We begin with a further study of the contribution of the coupled attention layer to the attention difference Equation 25, we now introduce the per layer linear operators

$$R_\ell = 2\, A_{\ell,0}\, J_{\ell,V}\,, \quad P_\ell(g)\, h_\ell = 2\big(\delta A_\ell^{(+)} + g\, \delta A_\ell^{(-)}\big) V_{\ell,0}\,, \tag{26}$$

and the gating functions

$$\rho(g) = \frac{1-g}{1+g}\,, \qquad \xi(g) = \frac{1}{1+g}\,, \tag{27}$$

so that the attention difference Equation 25 becomes

$$\Delta\, \mathrm{Attn}_{g,\ell} = \rho(g)\, R_\ell\, h_\ell + \xi(g)\, P_\ell(g)\, h_\ell + \mathcal{O}(\|h_\ell\|^2)\,. \tag{28}$$

The sequential residual structure of the DiT block Equation 13–Equation 14 means that the MLP acts on the post attention hidden state $\tilde{H}_\ell$, not on the original input $H_\ell$. Denoting the attention contribution to the perturbation mode $v$ by

$$\Delta\, \mathrm{Attn}_{g,\ell} = [\rho(g)\, R_\ell + \xi(g)\, P_\ell(g)]\, h_\ell + \mathcal{O}(\|h_\ell\|^2) \tag{29}$$

as in Equation 28, the MLP sees a shifted input whose difference component is $h_\ell + \Delta \, \text{Attn}_{g,\ell}$. Linearizing the MLP around the symmetric state, the full one block difference mode update is therefore

$$h_\ell^+ = (\mathbb{I}_\ell + J_{\ell,0}^{MLP})(\mathbb{I}_\ell + \rho(g)\, R_\ell + \xi(g)\, P_\ell(g))\, h_\ell + \mathcal{O}(\|h_\ell\|^2)\,, \tag{30}$$

where $J_{\ell,0}^{MLP}$ is the pointwise MLP Jacobian evaluated at the symmetric state, incorporating the adaptive gating vector $\beta_\ell$ and $\mathbb{I}$ is the per-layer identity matrix. Algebraically expanding the product yields

$$h_\ell^+ = K_g\, h_\ell + \mathcal{O}(\|h_\ell\|^2)\,, \tag{31}$$

$$K_g = \mathbb{I}_\ell + J_{\ell,0}^{MLP} + \rho(g)\, R_\ell + \xi(g)\, P_\ell(g) + J_{\ell,0}^{MLP}\big[\rho(g)\, R_\ell + \xi(g)\, P_\ell(g)\big]\,. \tag{32}$$

The cross term is first order in the perturbation amplitude $\|h_\ell\|$, it acts once on $h_\ell$ but is a product of two operators each set by the residual gating scale $\epsilon = \mathcal{O}(\|\alpha_\ell\|, \|\beta_\ell\|)$, since $J_{\ell,0}^{MLP} = \mathcal{O}(\beta_\ell)$ and $R_\ell, P_\ell(g) = \mathcal{O}(\alpha_\ell)$. These are two independent truncations: $\|h_\ell\|$ controls the linearization remainder $\mathcal{O}(\|h_\ell\|^2)$, whereas $\epsilon$ controls the cross term, which is $\mathcal{O}(\epsilon^2)$ against $\mathcal{O}(\epsilon)$ for the individual attention and MLP contributions. At initialization $\alpha_\ell, \beta_\ell = 0$, so the cross term vanishes and $K_g$ reduces to

$$K_g \approx \mathbb{I}_\ell + J_{\ell,0}^{MLP} + \rho(g)\, R_\ell + \xi(g)\, P_\ell(g)\,. \tag{33}$$

In the pretrained network, however, the gates are $\mathcal{O}(1)$ and grow with depth (RMS $\|\alpha_\ell\|, \|\beta_\ell\|$ up to 1.4–2.7; figure 9), and the cross term reaches $\approx 44\%$ of the first order block update figure 10. We therefore do not drop it and retain $K_g$ in the full form Equation 32. This leaves the predictions unchanged, the cross term factorizes as $\rho(g)\, J_{\ell,0}^{MLP} R_\ell + \xi(g)\, J_{\ell,0}^{MLP} P_\ell(g)$, carrying the same coupling gates $\rho(g), \xi(g)$ so it collapses with $g$ identically, inherits the routing operator's spatial structure from $R_\ell$, and is absorbed into the effective modal gains $\chi_k, \pi_k$ Equation 47 without altering the gating structure on which the gap collapse rests.

Around the unperturbed symmetric state $v = 0$, the LayerNorm Jacobian is well defined and is absorbed into the effective linear operators $R_\ell$, $P_\ell(g)$, and $J_{\ell,0}^{MLP}$ without altering the structure of the expansion Equation 25. Similarly, the adaptive residual gating vectors $\alpha_\ell$ and $\beta_\ell$ act as layer dependent, elementwise rescalings of the attention and MLP contributions, respectively. These vectors are absorbed into $R_\ell$, $P_\ell(g)$ (via $\alpha_\ell$) and $J_{\ell,0}^{MLP}$ (via $\beta_\ell$). Because these gating vectors are predicted from the conditioning embedding and are initialized to zero during training, they can introduce significant layer to layer variation in the effective propagator, which is one mechanism behind the depth localization. Consequently, $K_g$ in Equation 32 is the effective per layer Jacobian of the difference mode update around the symmetric state.

In this approximation, the MLP contribution $J_{\ell,0}^{MLP}$ acts independently on each spatial token $MLP(H)_n = \phi(H_n)$ for $n = 1, \ldots, N$, where $\phi$ is the same nonlinear function applied at every position. Therefore $J_{\ell,0}^{MLP}$ is block diagonal in token space and cannot perform explicit spatial routing, it processes the difference mode token by token without cross position information exchange. The retained cross term $J_{\ell,0}^{MLP}[\rho(g)\, R_\ell + \xi(g)\, P_\ell(g)]$ factorizes along the same gates into a routing based piece $\rho(g)\, J_{\ell,0}^{MLP} R_\ell$ and a modulation based piece $\xi(g)\, J_{\ell,0}^{MLP} P_\ell(g)$. Therefore, it preserves the channel decomposition, and is absorbed into the effective gains $\chi_k, \pi_k$ accordingly.

The linearized propagator $K_g$ Equation 31 describes how the perturbation is transported through the network, but a linear map around a single fixed point cannot produce speciation, it can only damp or amplify the difference (cf. discussion of §2.1). To derive a speciation criterion, we require a non-linear score with multiple attractors. Therefore, we model the local marginal distribution of the perturbation at reverse step $s$ and layer $\ell$ using the simplest nontrivial case which is a symmetric two component Gaussian mixture (Biroli, Bonnaire, Bortoli, and Mézard, 2024)[2]

$$p_{s,\ell}(h) = \tfrac{1}{2}\mathcal{N}\big(h; +m_{s,\ell},\, C_{s,\ell}\big) + \tfrac{1}{2}\mathcal{N}\big(h; -m_{s,\ell},\, C_{s,\ell}\big)\,, \tag{34}$$

---

[2]Equal covariances and two components are the simplest choice exhibiting a pitchfork bifurcation, and model a single binary speciation per mode. Unequal covariances would make the two branches asymmetric and shift the bifurcation threshold (a tilted, imperfect pitchfork) without removing it. Additional components would admit more than two basins per mode (the multi-class extension of (Achilli, Benedetti, Biroli, and Mézard, 2026)). Neither affects the leading/trailing ordering studied here, which depends only on the sign and monotonicity of the modewise speciation parameter, not on the number or symmetry of branches.

where $m_{s,\ell} \in R^D$ is the branch separation vector and $C_{s,\ell} \succ 0$ is the local strictly positive definite covariance matrix. For bookkeeping purposes, we will call objects at fixed step $s$ and layer $\ell$, a local object. The corresponding score is

$$\nabla_h \log p_{s,\ell}(h) = -C_{s,\ell}^{-1} h + C_{s,\ell}^{-1} m_{s,\ell} \tanh\!\left(m_{s,\ell}^\top C_{s,\ell}^{-1} h\right). \tag{35}$$

The first term is the linear restoring force, identical to the single Gaussian case. The second term provides the nonlinearity, the tanh saturates for large $|m^\top C^{-1} h|$, creating a cubic like bifurcation structure that admits nontrivial fixed points.

We model the deterministic one block update by combining the linearized propagator $K_g$ Equation 31 with a local score gain $\gamma_{s,\ell} > 0$ which is the discrete analogue of $\beta_t$ times the inverse noise variance in the continuous theory. In the pretrained DiT, it absorbs the combined effect of the AdaLN modulation and the learned output projection scale of each block. Local score gain is measurable from network activations by comparing the magnitude of the score like update to the residual stream. We estimate it through the residual update proxy $\gamma_{s,\ell} \propto \|H_{\ell+1} - H_\ell\|/\|H_\ell\|$ (section 4), which confirms the monotonicity, this is important for qualitative predictions such as the mode ordering, gap existence, and gap collapse. A first principles measurement of $\gamma$ calibration of its absolute scale we leave to future work. In total, we obtain

$$h^+ = K_g\, h + \gamma_{s,\ell}\left[-C_{s,\ell}^{-1} h + C_{s,\ell}^{-1} m_{s,\ell} \tanh\!\left(m_{s,\ell}^\top C_{s,\ell}^{-1} h\right)\right]. \tag{36}$$

The one block map Equation 36 combines two conceptually distinct contributions to the difference mode $v$ dynamics. The propagator Equation 31 captures the architectural response of the block, how self-attention routes the difference signal across spatial positions and how the pointwise MLP reshapes it. Because $K_g$ is obtained by linearizing the block around the replica symmetric point $v = 0$, where the mixture score Equation 35 itself vanishes, $K_g$ encodes the Jacobian of the block evaluated at the point of zero score.

The second term, proportional to $\gamma_{s,\ell}$, models the effective per block score increment, the fraction of the total score response attributable to block $\ell$ at reverse step $s$, evaluated on the two component mixture Equation 34. In a deep residual network with $L$ blocks, the full score is built up incrementally, $\gamma_{s,\ell}$ parameterizes each block's share.

A potential concern is that the linearized part of the score overlaps with the propagator $K_g$. To see this explicitly, expand Equation 35 around the difference mode $v = 0$

$$\nabla_v \log p_{s,\ell}(v) = -\Lambda_{\text{eff}}\, v + \mathcal{O}(\|v\|^3)\,, \tag{37}$$

where effective precision of the mixture is

$$\Lambda_{\text{eff}} = C_{s,\ell}^{-1} - C_{s,\ell}^{-1} m_{s,\ell}\, m_{s,\ell}^\top C_{s,\ell}^{-1}\,. \tag{38}$$

The first piece $-C_{s,\ell}^{-1} v$ is the single Gaussian restoring force and the rank one correction $C_{s,\ell}^{-1} m_{s,\ell}\, m_{s,\ell}^\top C_{s,\ell}^{-1} v$ is the linear contribution from expanding the tanh and encodes the partial cancellation of the restoring force along the branch separation direction.

One could therefore absorb all linear in $v$ score contributions into a redefined propagator

$$\tilde{K}_g = K_g - \gamma_{s,\ell}\, \Lambda_{\text{eff}}\,, \tag{39}$$

leaving only the genuinely nonlinear remainder

$$\gamma_{s,\ell}\, C_{s,\ell}^{-1} m_{s,\ell}\left[\tanh\!\left(m_{s,\ell}^\top C_{s,\ell}^{-1} v\right) - m_{s,\ell}^\top C_{s,\ell}^{-1} v\right] = \mathcal{O}(\|v\|^3)\,, \tag{40}$$

as the only score like term beyond the propagator. This repartitioning leaves the fixed point Equation 41 and the scalar self consistency condition Equation 49 exactly invariant. Both conditions are derived from the full tanh nonlinearity, not from the linearization, so they already correctly account for the rank one linear piece.

This decomposition is analogous to separating the free propagator from the self energy in a diagrammatic expansion in quantum field theory (Zinn-Justin, 2021). In that context, $K_g$ plays the role of the bare

propagator, $-\gamma_{s,\ell}\,\Lambda_{\text{eff}}$ is the one loop self energy correction from the data distribution, and the tanh remainder Equation 40 contains the higher order vertices. We treat $\gamma_{s,\ell}$ as an effective coupling, it absorbs any overlap between the architectural and score like linear contributions, so that the physically meaningful and measurable quantity is the total effective Jacobian at the symmetric point, $K_g - \gamma_{s,\ell}\,\Lambda_{\text{eff}}$, not the individual partition. All testable predictions depend on this total Jacobian and on the bifurcation structure of the tanh term, they are therefore invariant under repartitioning between $K_g$ and $\gamma_{s,\ell}$.

A fixed point satisfies

$$\left[(\mathbb{I} - K_g) + \gamma_{s,\ell}\,C_{s,\ell}^{-1}\right] v = \gamma_{s,\ell}\,C_{s,\ell}^{-1}\,m_{s,\ell}\,\tanh\!\big(m_{s,\ell}^{\top}C_{s,\ell}^{-1}\,v\big)\,. \tag{41}$$

This is the discrete, attention gated analogue of the continuous fixed point in the symmetric coupled OU analysis of Albrychiewicz et al. (2026). We emphasize that the condition $v^{+} = v$ should not be read as a literal dynamical equilibrium of a single transformer block. Here, the difference mode flows through all blocks and reverse steps without equilibrating at any one of them. Rather, Equation 41 is a local stability criterion, at each $(s,\ell)$ it asks whether the effective one block potential for $v$ has developed a bifurcation point. This bifurcation point is where the symmetric solution $v = 0$ becomes locally unstable and two new attracting directions appear, signaling the onset of branch formation. This is the standard interpretation of the self consistency equation in mean field theory (Mezard and Montanari, 2009) of phase transitions (cf., the Curie–Weiss model (Kittel and McEuen, 2018)), applied here to the discrete computational graph of the transformer.

## 2.5 Modewise Signal-to-Noise Ratio Formula

In this subsection, we project the one block fixed point Equation 41 onto the empirical difference modes and, under a mean field single mode approximation, reduce it to one scalar self consistency equation per mode Equation 49. Its pitchfork bifurcation defines the modewise speciation parameter Equation 50, equivalently an attention gated signal-to-noise ratio (SNR) Equation 53, whose dependence on the routing gain $\chi_k$ is what makes speciation frequency selective. We start with the projection onto fixed empirical modes and determine their bifurcation. Let

$$C_{0,\ell}^{\text{emp}} = \frac{1}{M}V_{0,\ell}^{\top}V_{0,\ell} \in \mathbb{R}^{D \times D} \tag{42}$$

denote the empirical covariance of the hidden state difference vectors at the initial reverse step $s = 0$ ($t = T$) and layer $\ell$, where $V_{0,\ell} \in \mathbb{R}^{M \times D}$ stacks the $M$ sampled difference vectors row wise and $D$ is the dimension of each difference vector at layer $\ell$. Let $\{r_k^{(\ell)}\}_{k=1}^{K}$ be its leading eigenvectors, obtained numerically through the dual Gram matrix

$$G_{0,\ell} = \frac{1}{M}V_{0,\ell}V_{0,\ell}^{\top} \in \mathbb{R}^{M \times M} \tag{43}$$

via the Nyström construction. For each reverse step $s$ and layer $\ell$, define the modal projections

$$c_k = r_k^{\top}C_{s,\ell}\,r_k\,, \qquad m_k = r_k^{\top}m_{s,\ell}\,, \qquad \eta_k(g) = r_k^{\top}K_g\,r_k\,, \tag{44}$$

here $c_k$ is a projected covariance, and $\eta_k(g)$ is a projected one block gain and we suppress the $(s,\ell)$ subscripts at left hand side for clarity.

In a mean field approximation one replaces the coupled multi mode fixed point by independent scalar self consistency equations, one per mode, obtained by projecting onto a single dominant direction and neglecting inter mode coupling Mézard et al. (1987). Therefore, we set $v = a_k\,r_k$ and derive a scalar self consistency equation for each mode independently, we refer to this as single mode ansatz. This decoupling is justified under two conditions. First, the empirical modes $\{r_k\}$ are taken to be approximately orthonormal eigenvectors of the difference covariance $C_{s,\ell}$. By construction Equation 43 they are the eigenvectors of the symmetric positive semidefinite empirical covariance matrix $C_{0,\ell}^{\text{emp}}$ at the initial step $s = 0$, hence exactly orthonormal there by the spectral theorem. For $s > 0$ the covariance $C_{s,\ell}$ drifts, so $\{r_k\}$ diagonalize it only approximately. Nonetheless, we retain the $s = 0$ eigenbasis as a fixed coordinate system throughout (as in

Protocol II, section 3.2), which is what makes the modal projections Equation 44 a stable, $s$-independent basis rather than an $s$-dependent one. Second, the branch separation vector $m_{s,\ell}$ in the mixture model Equation 34 is assumed to lie predominantly in span$\{r_k\}$, i.e. the replicas separate along the high variance difference directions. This is expected because $m_{s,\ell}$ is the direction along which the two mixture components separate, and that separation is where difference channel energy concentrates. A component of $m_{s,\ell}$ orthogonal to the leading modes would carry separation energy invisible to the modal projection Equation 44, and the scalar reduction would underestimate the corresponding bifurcation. Empirically the leading modes capture the bulk of the difference energy at the speciation time (the fixed basis energies of section 4), which supports the assumption, otherwise, the single mode ordering would remain valid only for the captured energy fraction.

Under the single mode ansatz, the argument of the tanh nonlinearity collapses to a scalar

$$m_{s,\ell}^\top C_{s,\ell}^{-1} v = a_k \frac{m_k}{c_k} = u_k \,, \tag{45}$$

where we define $u_k$ as the rescaled modal order parameter. If $v$ instead contained a superposition of non-independent modes, the tanh argument would become a sum $\sum_j u_j$, and the nonlinearity would generate cross mode couplings. However, because the tanh expansion contains no even powers, these nonlinear cross mode corrections enter strictly at $\mathcal{O}(\|v\|^3)$. Furthermore, any linear mode mixing arises only from the off diagonal elements of $K_g$ in the $\{r_k\}$ basis. In practice, since $\{r_k\}$ span the dominant variance directions at initialization, these off diagonal terms are small. Replacing the coupled multi mode fixed point Equation 41 by one scalar equation per mode amounts to neglecting all inter-mode coupling. That coupling has only two sources, and both are controlled near onset. The nonlinear source vanishes to leading order because the tanh is odd, so cross mode terms enter at $\mathcal{O}(\|v\|^3)$. The linear source is the off diagonal of $K_g$ in the $\{r_k\}$ basis, which is small precisely because $\{r_k\}$ are the dominant variance directions at initialization, so $K_g$ is nearly diagonal there. Consequently, near the symmetric bifurcation point where $|u_k|$ is small the diagonal (single mode) closure is accurate, and we treat the onset of instability for the leading modes independently. The approximation is expected to degrade away from criticality, where the $\mathcal{O}(\|v\|^3)$ cross-mode terms are no longer subleading.

Using the empirical spectral decomposition, we expand $\eta_k(g)$ with Equation 31

$$\eta_k(g) = 1 + \lambda_k^{MLP} + \rho(g)\,\chi_k + \xi(g)\,\pi_k \,, \tag{46}$$

where

$$\lambda_k^{MLP} = r_k^\top J_{MLP}\, r_k \,, \qquad \chi_k = r_k^\top R\, r_k \,, \qquad \pi_k = r_k^\top P_g\, r_k \,. \tag{47}$$

Here, $\lambda_k^{MLP}$ is the pointwise MLP contribution, $\chi_k$ the modewise spatial routing gain, the quantity that is frequency selective by virtue of the learned attention patterns $A_0$ and $\pi_k$ the pattern modulation correction. Retaining the cross term of Equation 32 renormalizes these to $\chi_k = r_k^\top(\mathbb{I} + J^{MLP})R\, r_k$ and $\pi_k = r_k^\top(\mathbb{I} + J^{MLP})P_g\, r_k$, leaving the form of Equation 46 and the $\rho(g), \xi(g)$ gating unchanged. Since $J^{MLP}$ is nearly mode independent, $(\mathbb{I} + J^{MLP})R$ retains the low pass structure of $R$. Both gains are quadratic forms in the modes $r_k$ whose kernels are built from $A_{\ell,0}$, linearly through $R$, and through $P_g$ via the softmax Jacobian $\partial A/\partial S = \mathrm{diag}(A_0) - A_0 A_0^\top$ entering $\delta A^{(\pm)}$, and therefore depend on the point at which the softmax is linearized. A peaked $A_{\ell,0}$ yields a worse conditioned Jacobian than a diffuse one, the size of peak is inversely proportional to per row effective width $N_{\mathrm{eff}}^{(i)}$ of Appendix B. This dependence is not negligible, rather than bound it, we measure the routing and modulation channels directly through the energy fractions (section 4.3) and the spatial low pass profile of $A_{\ell,0}$ (Appendix B) that sets their mode dependence, so the conditioning enters our predictions through their measured values. The routing/modulation split itself, Equation 25 is independent of it, following from the exact row stochastic identities $\delta A\,\mathbf{1} = 0$ and $A_0 P_0 = P_0$.

With the mode expansion and assuming that $C_{s,\ell}$ is diagonal in $r_k$ basis, we can rewrite fixed point as Equation 41

$$\left[(1 - \eta_k(g)) + \frac{\gamma_{s,\ell}}{c_k}\right] a_k = \frac{\gamma_{s,\ell}\, m_k}{c_k}\, \tanh\!\left(\frac{m_k}{c_k}\, a_k\right). \tag{48}$$

In terms of $u_k$, this simplifies to

$$u_k = \kappa_{v,k}(s, \ell;\, g)\, \tanh(u_k)\,, \tag{49}$$

with the modewise speciation parameter

$$\kappa_{v,k}(s, \ell;\, g) = \frac{\gamma_{s,\ell}\, m_k^2}{c_k\Big((1 - \eta_k(g))\, c_k + \gamma_{s,\ell}\Big)}\,. \tag{50}$$

Equation 49 has the standard mean field bifurcation structure for $\kappa_{v,k} \leq 1$, and the only solution is $u_k = 0$ given when replicas are identical. For $\kappa_{v,k} > 1$, two additional nonzero solutions emerge symmetrically, so replicas are committed to distinct branches. The modal speciation step is therefore defined by

$$\kappa_{v,k}\big(s_{spec}^{(k)},\, \ell;\, g\big) = 1\,, \tag{51}$$

which is when the equation undergoes a standard pitchfork bifurcation.

Following the logic of Albrychiewicz et al. (2026), we factor the speciation parameter into a score gain and a signal-to-noise ratio

$$\kappa_{v,k}(s, \ell;\, g) = \gamma_{s,\ell}\, SNR_{v,k}(s, \ell;\, g)\,, \tag{52}$$

where the modewise difference mode SNR is

$$SNR_{v,k}(s, \ell;\, g) = \frac{m_k^2}{c_k\Big((1 - \eta_k(g))\, c_k + \gamma_{s,\ell}\Big)}\,. \tag{53}$$

Substituting the decomposition Equation 46 and writing $\mu_k = 1/c_k$ for the modal precision, this expands to

$$SNR_{v,k}(s, \ell;\, g) = \frac{m_k^2\, \mu_k^2}{\gamma_{s,\ell}\mu_k - \lambda_k^{MLP} - \rho(g)\, \chi_k - \xi(g)\, \pi_k}\,. \tag{54}$$

The SNR formula Equation 54 is well defined and positive, provided that the denominator is strictly positive:

$$\gamma_{s,\ell}\, \mu_k > \lambda_k^{MLP} + \rho(g)\, \chi_k + \xi(g)\, \pi_k\,, \tag{55}$$

or can be equivalently written as $\eta_k(g) < 1 + \gamma_{s,\ell}/c_k$. This condition states that the combined score gain and modal precision must exceed the per block amplification of the difference mode. In the opposite regime, $\eta_k(g) \geq 1 + \gamma_{s,\ell}/c_k$, the linearized difference mode is amplified faster than the score can restore it. Hence, the mean field picture breaks down, and a non perturbative treatment would be required.

To obtain an analytic theory, we adopt the phenomenological ansatz that the branch separation amplitude propagates multiplicatively along the trajectory,

$$m_{k,s,\ell} = G_{v,k}(s, \ell;\, g)\, m_{k,\text{init}}. \tag{56}$$

This equation on its own only defines $G_{v,k} = m_{k,s,\ell}/m_{k,\text{init}}$. Here, the assumption we make is the specific form of $G_{v,k}$,

$$G_{v,k}(s, \ell;\, g) = \prod_{(s',\ell') \prec (s,\ell)} \eta_{k,s',\ell'}(g)\,, \tag{57}$$

where the ordered pair $(s', \ell')$ indexes the reverse step $s'$ and transformer layer $\ell'$, and the relation $(s', \ell') \prec (s, \ell)$ means that the block at $(s', \ell')$ is processed earlier in the reverse trajectory than the block at $(s, \ell)$. Namely, the branch amplitude is transported by the running product of the same per step modal gains $\eta_k$ that govern the linearized propagator with no additional mode dependent source. However, the closure fails if $m_k$ grows by a mechanism the propagator does not capture, such as cross mode feeding (excluded here by the diagonal approximation above) or a nonlinear injection beyond the tanh saturation in which case

$G_{v,k}$ would deviate from $\prod \eta_k$. It is testable because the propagated SNR Equation 58 below depends on $G_{v,k}^2 = \prod \eta_k^2$, the same cumulative squared gain measured directly by the fixed basis mode energies $\lambda_k(s, \ell)$ (Equation 75). The agreement of these energy trajectories with the predicted leading/trailing ordering section 4 is the empirical check on Equation 57, not on Equation 56.

Now, we can rewrite the propagated SNR in terms of $G_{v,k}$ as

$$SNR_{v,k}(s, \ell; g) = \frac{G_{v,k}(s, \ell; g)^2 \, m_{k,\text{init}}^2}{c_{k,s,\ell}\Big((1 - \eta_{k,s,\ell}(g)) \, c_{k,s,\ell} + \gamma_{s,\ell}\Big)} \, . \tag{58}$$

This is the discrete counterpart of the continuous SNR of the coupled OU analysis of Albrychiewicz et al. (2026). The OU processes eigenvalues are replaced by the learned attention routing gains $\chi_k$, and the coupling enters through $\rho(g)$, $\xi(g)$ and the cumulative discrete gain $G_{v,k}^2(g)$. Heuristically, one may view the layerwise propagation of the difference mode as resembling a discrete inverse renormalization group flow. Here, discrete inverse renormalization group (RG) denotes the reversal of RG coarse graining, where a forward RG step locally averages out high frequency detail and flows the effective couplings from fine to coarse scales, run over the discrete scale axis. An example of RG flow application to diffusion models can be found in Masuki & Ashida (2025). In this analogy, the reverse step $s$ and layer depth $\ell$ play the role of a discrete scale parameter, while the learned attention map $A_0$ acts as a data dependent spatial averaging operator. In this framework, the cumulative gain Equation 56 can be interpreted as representing the discrete flow of the modal operators, and the mode wise branch separation $m_k$ evolves according to the propagator $\eta_k(g)$. We stress, however, that in this paper, this is only an interpretive analogy, no exact RG identification is used or required for the derivations above.

## 2.6 The Synchronization Gap and Its Collapse

We will now focus on deriving the central experimental observable that we are interested in measuring i.e., the synchronization gap between leading and trailing difference modes. To this end, we combine the modewise SNR Equation 53 with the routing term dominance regime of Appendix B, in which attention routing dominates the pattern modulation correction and favors coarse over fine modes. We compare the speciation times of a coarse and a fine mode, we show the coarse mode commits first, define the resulting synchronization gap Equation 65, and establish that it scales as $\mathcal{O}\left(\frac{1-g}{1+g}\right)$ Equation 66, hence collapsing as $g \to 1$.

Let $k_{\text{hi}}$ denote a leading mode associated with coarse, globally organized structure and $k_{\text{lo}}$ a trailing mode associated with finer detail. We order modes by their descending eigenvalues of the initial difference covariance $C_{0,\ell}^{\text{emp}}$: $k_{\text{hi}}$ is a leading mode (largest variance) and $k_{\text{lo}}$ a trailing mode (smallest variance). The identification of leading modes with long wavelength (low spatial frequency) is empirical, which is the measured low pass profile of $A_0$ (section 4), not a convention. We note that this is opposite to the graph Laplacian ordering, where a low index denotes low frequency. We use leading/trailing throughout for this internal eigenmode ordering (Protocol II, section 3.2) and reserve global/local for the output space scale decomposition (Protocol I, section 3.1). The correspondence leading↔global, trailing↔local is the result of empirical low pass identification above.

In the early denoising steps, low frequency regime characterized in Appendix B, we assume the following structural properties. First, the spatial routing contribution dominates the pattern-modulation correction,

$$|\pi_k| \ll |\chi_k| . \tag{59}$$

We provide a structural reason why this is valid in Appendix B. Next, the MLP term is only weakly mode dependent as we discussed in section 2.4

$$\lambda_k^{MLP} \approx \lambda^{MLP} \quad . \tag{60}$$

Finally, as a modeling hypothesis consistent with the routing dominant regime, we assume that attention routes leading coarse modes more strongly than trailing fine modes,

$$\chi_{k_{\text{hi}}} > \chi_{k_{\text{lo}}}. \tag{61}$$

From RG perspective, this can be interpreted as a coarse graining operator preserving $k_{\mathrm{hi}}$ (long wavelength) while heavily suppresses $k_{\mathrm{lo}}$ (respectively short wavelength). Under these structural properties, the propagator Equation 46 simplifies to

$$\eta_k(g) \approx 1 + \lambda^{MLP} + \rho(g)\,\chi_k\,, \tag{62}$$

and the routing dominant SNR becomes (cf., Equation 53)

$$SNR_{v,k}(s,\ell;g) \approx \frac{m_k^2}{c_k\left(\left(-\lambda^{MLP} - \rho(g)\,\chi_k\right)c_k + \gamma_{s,\ell}\right)}\,. \tag{63}$$

If the remaining modal factors $(m_k, c_k)$ are comparable, then $SNR_{v,k_{\mathrm{hi}}} > SNR_{v,k_{\mathrm{lo}}}$, which implies

$$s_{spec}^{(k_{\mathrm{hi}})}(\ell;g) < s_{spec}^{(k_{\mathrm{lo}})}(\ell;g), \tag{64}$$

the leading mode speciates before the trailing mode. The modewise synchronization gap at layer $\ell$ and coupling $g$ is

$$\Delta s_v(\ell;g) := s_{spec}^{(k_{\mathrm{lo}})}(\ell;g) - s_{spec}^{(k_{\mathrm{hi}})}(\ell;g)\,. \tag{65}$$

A positive value indicates that the leading mode has committed while the trailing mode remains unresolved. Using the assumptions from above, the SNR difference between modes picks up a factor of $\rho(g)$ from the routing term. To the first order in the spectral split, we have

$$SNR_{v,k_{\mathrm{hi}}}(s,\ell;g) - SNR_{v,k_{\mathrm{lo}}}(s,\ell;g) = \mathcal{O}\left(\frac{1-g}{1+g}\right)\,. \tag{66}$$

Hence, the spatial routing contribution to the SNR difference vanishes as $g \to 1$.

Two interpretive points follow. $g \to 1$ gates the routing channel off in Equation 66, which is precisely why the leading/trailing hierarchy and hence the gap collapses. $g = 0$ leaves routing fully on, so the gap measured there reflects the network's intrinsic spatial routing. The practical analogue of $g \to 1$ is coupled or shared-noise sampling that forces two generations to agree, deliberately suppressing difference channel diversity, whereas, $g = 0$ is the ordinary independent sampling regime that every standard DiT run already realizes. In consequence, a nonzero gap means the network fixes a sample's coarse layout several reverse steps before its fine detail, and resolves fine detail last in the deepest blocks. When there is no gap, all spatial scales would commit on a single timescale.

We intend to use this framework as an effective theory of the actual pretrained DiT, not as a free standing toy model. Its components are derived from the architecture and then closed by quantities measured on the network, so each assumption is a checkable statement. Concretely, the backbone of the derivation is exact, the routing/modulation split Equation 25 and the difference channel identities of Appendix B follow from row stochasticity alone and hold for any $A_{\ell,0}$. The remaining are now measured rather than assumed. (i) Routing dominance $|\pi_k| \ll |\chi_k|$ is measured directly, the routing and modulation fractions of the attention difference Equation 25 are $r_{\mathrm{route}} \approx 0.95$ versus $r_{\mathrm{mod}} \approx 0.27$ at $g = 0$ in the early reverse phase (section 4). (ii) Frequency selectivity $\chi_{k_{\mathrm{hi}}} > \chi_{k_{\mathrm{lo}}}$ is measured as the spatial low pass retention of $A_{\ell,0}$ on the difference modes, which falls monotonically with mode frequency (correlation $-0.95$; Appendix B, section 4). (iii) The local score gain $\gamma_{s,\ell}$ is estimated through a DiT block residual update $\gamma_{s,\ell} \propto \|H_{\ell+1} - H_\ell\|/\|H_\ell\|$ (median $\approx 0.17$ across $(\ell, s)$, increasing toward the late blocks; section 4), confirming the monotonicity on which the predictions depend. The only approximations that remain are: the single mode ansatz (cf., section 2.5), valid near the bifurcation and degrading away from it, and the assumption of weak mode dependence (same dependence for every mode $k$) of $\lambda_k^{MLP}$ and $m_k, c_k$, which affect only the ordering of speciation times, not the existence of the gap.

## 2.7 Summary of Testable Predictions

The derived theoretical framework yields four concrete empirical predictions.

1. **Commitment Hierarchy:** Leading modes speciate before trailing modes, $s_{spec}^{(k_{\mathrm{hi}})} < s_{spec}^{(k_{\mathrm{lo}})}$ (Equation 51), at the output level this appears as $\tau_{\mathrm{g}} < \tau_{\mathrm{l}}$ (Equation 73). The speciation time $\tau_{\mathrm{spec}}$ tracks this transition but is not required by the theory to lie strictly between them.

2. **Depth Localization:** The modal gain $\eta_k(g)$ Equation 46 and the score gain $\gamma_{s,\ell}$ inherit the layer dependent residual gates $\alpha_\ell, \beta_\ell$ of the DiT block Equation 13- Equation 14, so the speciation criterion Equation 51 depends on $\ell$ and the gap is not uniform across depth. Because these gates are empirically largest in the late blocks (section 4), the gap is predicted to concentrate there.

3. **Existence of the Natural Gap:** At $g = 0$, fully decoupled replicas sharing an initialization $z_T$ exhibit an intrinsic synchronization gap whose magnitude and mode dependence are predicted by the routing dominant SNR formula Equation 63.

4. **Coupling Induced Collapse:** As symmetric coupling $g$ increases toward 1, the spatial routing difference is suppressed, forcing both the internal layerwise gap and the behavioral commitment gap to collapse.

## 3 Empirical Setup

In this section, we describe experimental setup to verify theoretical predictions made in the previous section.

We begin with variance preserving initialization to ensure that the difference mode $v$ measurement is not confounded by pushing it out of prior distribution. We initialize the replicas with an antisymmetric perturbation that preserves the marginal variance of the generation prior probability

$$z_T \sim \mathcal{N}(0, \mathbb{I}), \quad \delta \sim \mathcal{N}(0, \mathbb{I}), \quad z_T^{A,B} = \frac{z_T \pm \sigma\, \delta}{\sqrt{1 + \sigma^2}}. \tag{67}$$

This guarantees $\mathrm{Var}(z_T^A) = \mathrm{Var}(z_T^B) = \mathbb{I}$ while injecting a controlled difference signal

$$v_T = \sqrt{2}\, \sigma\, \delta / \sqrt{1 + \sigma^2}. \tag{68}$$

The inter replica correlation is

$$\varrho(\sigma) = \frac{1 - \sigma^2}{1 + \sigma^2}, \tag{69}$$

for the experimentally relevant range $0 \le \sigma \le 1$, this interpolates from identical replicas $\varrho = 1$ to independent replicas $\varrho = 0$. For $\sigma > 1$, the replicas become negatively correlated.

### 3.1 Protocol I: Speciation Time and Scale Dependent Commitment

Our first empirical protocol probes the speciation time Equation 51 of the generative model. Two replicas are initialized with a shared macroscopic structure, a shared initial latent state $z_T$ and an antisymmetric noise perturbation, and coupled with strength $g$ up to an intervention step $t_{\mathrm{int}}$, after which they evolve independently $g = 0$ using Denoising Diffusion Implicit Model (DDIM) sampler (Song, Meng, and Ermon, 2020a) with $\eta = 1$ effectively recovering Denoising Diffusion Probabilistic Model (DDPM) (Ho, Jain, and Abbeel, 2020). By measuring the divergence of the final decoded outputs $x^A(t_{\mathrm{int}}; g)$ and $x^B(t_{\mathrm{int}}; g)$, we measure the time when bifurcation occurs.

We measure output agreement with two methods: within class perceptual speciation and scale dependent output commitment. To determine when the stochastic DDPM sampler commits the trajectories to the same semantic basin of attraction, we measure the feature space cosine similarity of the final images using a pretrained ImageNet encoder (ResNet-50) (He, Zhang, Ren, and Sun, 2016).

$$a_{\mathrm{feat}}(t_{\mathrm{int}}; g) = \cos\Big(\phi(x^A(t_{\mathrm{int}}; g)),\, \phi(x^B(t_{\mathrm{int}}; g))\Big). \tag{70}$$

The sigmoid is fit to the median agreement curve aggregated over the $M$ paired seeds, where $\tau_{\mathrm{spec}}(g)$ is its inflection point, and we report 95% bootstrap confidence intervals over seeds.

To isolate how spatial routing resolves generative ambiguity, we decompose the image discrepancy into a coarse component and a residual fine component. Let $P$ denote adaptive average pooling to a fixed low resolution, and let $U$ denote bilinear upsampling back to the original image size. We then define

$$d_{\text{low}}(t_{\text{int}}; g) = \|P(x_0^A) - P(x_0^B)\|_2^2, \tag{71}$$

$$d_{\text{high}}(t_{\text{int}}; g) = \|(x_0^A - UP(x_0^A)) - (x_0^B - UP(x_0^B))\|_2^2. \tag{72}$$

Thus, $d_{\text{low}}$ measures discrepancy after strong spatial averaging; whereas, $d_{\text{high}}$ measures discrepancy in the corresponding high frequency residual. We extract the macroscopic $\tau_{\text{g}}$ and microscopic $\tau_{\text{l}}$ commitment times from the 50% thresholds of their respective sigmoid fits. The output space synchronization gap is directly quantified as

$$\Delta\tau(g) = \tau_{\text{l}}(g) - \tau_{\text{g}}(g). \tag{73}$$

A positive gap formally indicates that the continuous physics of the DiT resolves low frequency global structure before high frequency local detail. Here, the terms "global" and "local" refer specifically to this output space scale decomposition, and are distinct from the internal hidden state basis defined in Protocol II.

## 3.2 Protocol II: Internal Mode Stabilization and the Layerwise Gap

Our second empirical protocol probes how the continuous synchronization gap is represented internally across the discrete Transformer depth $\ell$. In this setup, the replicas remain coupled at a constant $g$ for the entire reverse trajectory.

At each step $s$ and layer $\ell$, we extract the flattened hidden state difference vector for $M$ paired seeds

$$V_{s,\ell} = \frac{1}{\sqrt{2}} \begin{bmatrix} \text{vec}(H_{s,\ell}^{A,(1)} - H_{s,\ell}^{B,(1)})^\top \\ \vdots \\ \text{vec}(H_{s,\ell}^{A,(M)} - H_{s,\ell}^{B,(M)})^\top \end{bmatrix} \in \mathbb{R}^{M \times D}. \tag{74}$$

To construct a stable fixed basis of empirical principal modes without rank-deficiency artifacts, we form the $M \times M$ dual Gram matrix Equation 43 as discussed in section 2.5. The normalized energy of mode $k$ over time is

$$\lambda_k(s, \ell) = \frac{\|V_{s,\ell} r_k^{(\ell)}\|^2}{\|V_{0,\ell} r_k^{(\ell)}\|^2}. \tag{75}$$

Equivalently,

$$\lambda_k(s, \ell) = \frac{r_k^{(\ell)\top} C_{s,\ell}^{\text{emp}} r_k^{(\ell)}}{\lambda_k^{(0,\ell)}}, \tag{76}$$

where $\lambda_k^{(0,\ell)}$ is an eigenvalue of initial empirical covariance $C_{0,\ell}^{\text{emp}}$. If $r_k^{(\ell)}$ also diagonalizes $C_{s,\ell}^{\text{emp}}$, these are eigenvalues of the current covariance otherwise these are fixed basis internal modes defined at $s = 0$. They provide a stable internal coordinate system to ask how leading and trailing difference channel $v$ directions behave over time.

We then evaluate these normalized energies at the estimated speciation time from Protocol I for particular value of $g$

$$s = \tau_{\text{spec}}(g). \tag{77}$$

Table 1: Summary of speciation times determined using the cosine similarity method and corresponding confidence intervals versus coupling strength $g$.

| Coupling strength $g$ | $\tau(g)$ [95% CI] |
|---|---|
| 0.1 | 63.6 [50.0, 96.6] |
| 0.3 | 48.0 [33.7, 85.3] |
| 0.5 | 46.3 [31.6, 82.6] |
| 0.7 | 45.6 [35.0, 59.7] |
| 0.9 | 45.0 [35.8, 56.0] |
| 1.0 | 44.2 [35.6, 58.8] |

For robustness, we aggregate clusters rather than relying on single indices

$$\bar{\lambda}_{\text{lead}}(\ell; g) = \frac{1}{|\mathcal{K}_{\text{lead}}|} \sum_{k \in \mathcal{K}_{\text{lead}}} \lambda_k\big(\tau_{\text{spec}}(g), \ell\big), \tag{78}$$

$$\bar{\lambda}_{\text{trail}}(\ell; g) = \frac{1}{|\mathcal{K}_{\text{trail}}|} \sum_{k \in \mathcal{K}_{\text{trail}}} \lambda_k\big(\tau_{\text{spec}}(g), \ell\big), \tag{79}$$

where $\mathcal{K}_{\text{lead}}(\ell)$ and $\mathcal{K}_{\text{trail}}(\ell)$ denote leading and trailing bands in the ordering induced by the initial covariance eigenvalues $\lambda_k^{(0,\ell)}$. This yields the internal energy gap ratio

$$\mathcal{G}_{\text{int}}(\ell; g) = \frac{\bar{\lambda}_{\text{trail}}(\ell; g)}{\bar{\lambda}_{\text{lead}}(\ell; g)}. \tag{80}$$

Values $\mathcal{G}_{\text{int}}(\ell; g) > 1$ indicate that trailing empirical modes retain more energy than leading empirical modes at the moment of the species commitment.

## 4 Results

In this section, we report the results of experiments described in the previous section. First, we determine the speciation time for a range of coupling strengths $g$ using the Protocol I experiment setup section 3.1. Second, we conduct a layer wise sweep to analyze the synchronization gap and its sensitivity to g as defined in Protocol II section 3.2. The two protocols use different samplers by design: Protocol I uses stochastic sampling ($\eta = 1$) because basin commitment is a property of the noisy reverse process, whereas Protocol II uses deterministic sampling ($\eta = 0$) to read internal mode energies along a single, noise free trajectory without sampling induced variance.

### 4.1 Protocol I Results

As mentioned before, we use DDIM (Song, Meng, and Ermon, 2020a) reverse sampler for DiT-XL/2 model. For Protocol I experiment, we set the noise parameter $\eta = 1$ which corresponds to stochastic DDPM sampler (Ho, Jain, and Abbeel, 2020). We apply the coupling for all transformer blocks and we apply intra/inter replica blocks normalization as shown in Equation 19. For each $g$, we sweep 100 reverse steps and we measure the cosine similarity of the final images in feature space using a pretrained ResNet-50 encoder. We use same ImageNet class id (0) in all experiments, but we also verify that behavior does not change when we pick different classes. We fit the sigmoid function to the median curve over $M = 32$ seeds and define the speciation time as a midpoint, and we also report the 95% bootstrap confidence interval. Finally, we show results for $g \in \{0.1, 0.3, 0.5, 0.7, 0.9, 1.0\}$ on figure 2a–figure 2f and also summarize them in Table 1 for readers' convenience.

Qualitatively, all curves exhibit the same structure predicted by the theory for small $t_{\text{int}}$, the paired trajectories remain close to the independently initialized baseline; while for sufficiently long shared evolution,

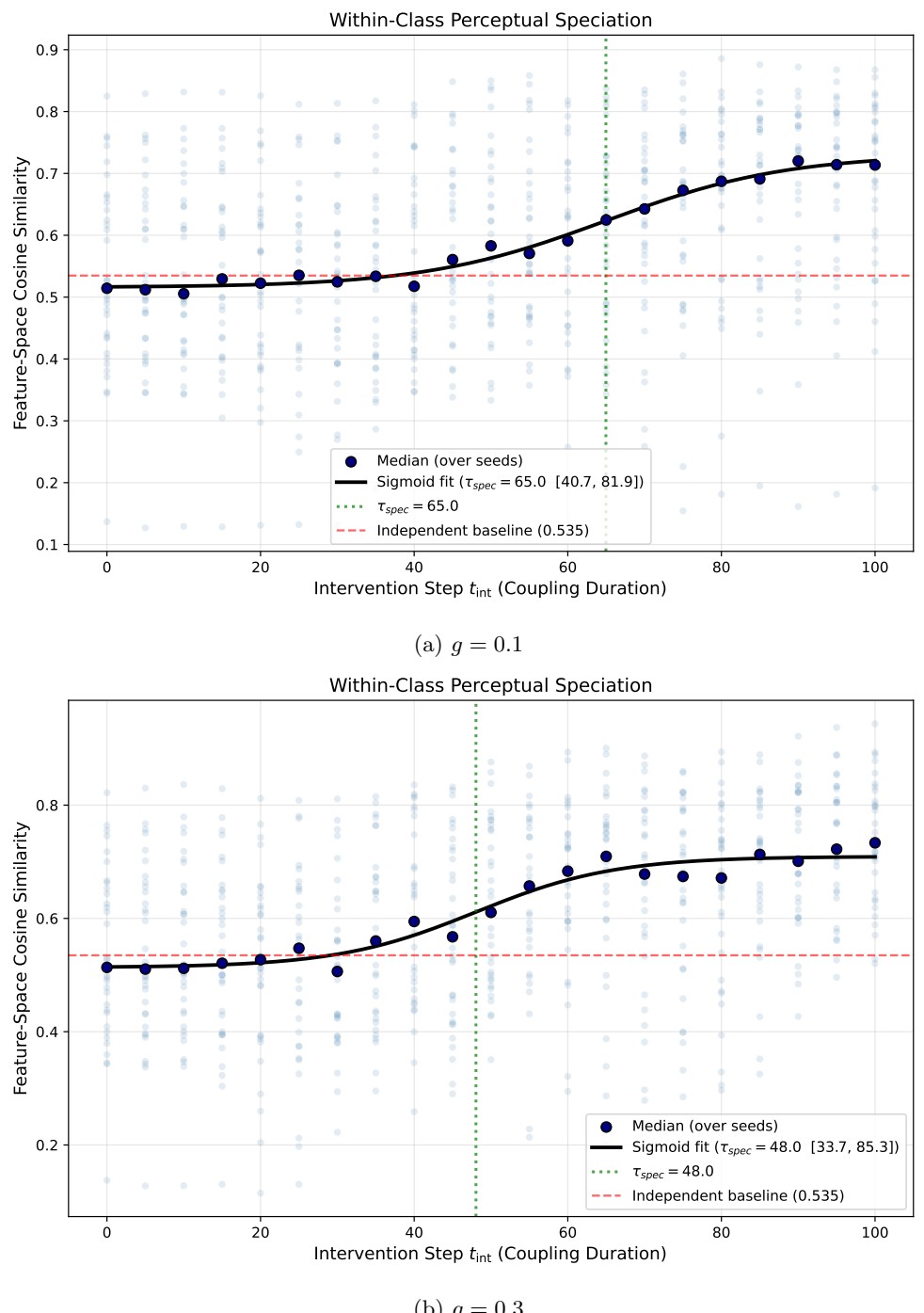

(a) $g = 0.1$

(b) $g = 0.3$

Figure 2: Each plot shows the median feature space cosine similarity Equation 70 between paired outputs as a function of intervention step $t_{\text{int}}$, together with the sigmoid fit used to extract $\tau_{\text{spec}}(g)$. With an increase of coupling strength $g$, the speciation time decreases.

the feature space agreement rises and saturates at a substantially higher level. The main coupling effect is therefore not a change in the final plateau itself, but an earlier step shift of the transition. This is consistent with the interpretation that stronger symmetric coupling suppresses replica difference spatial routing earlier

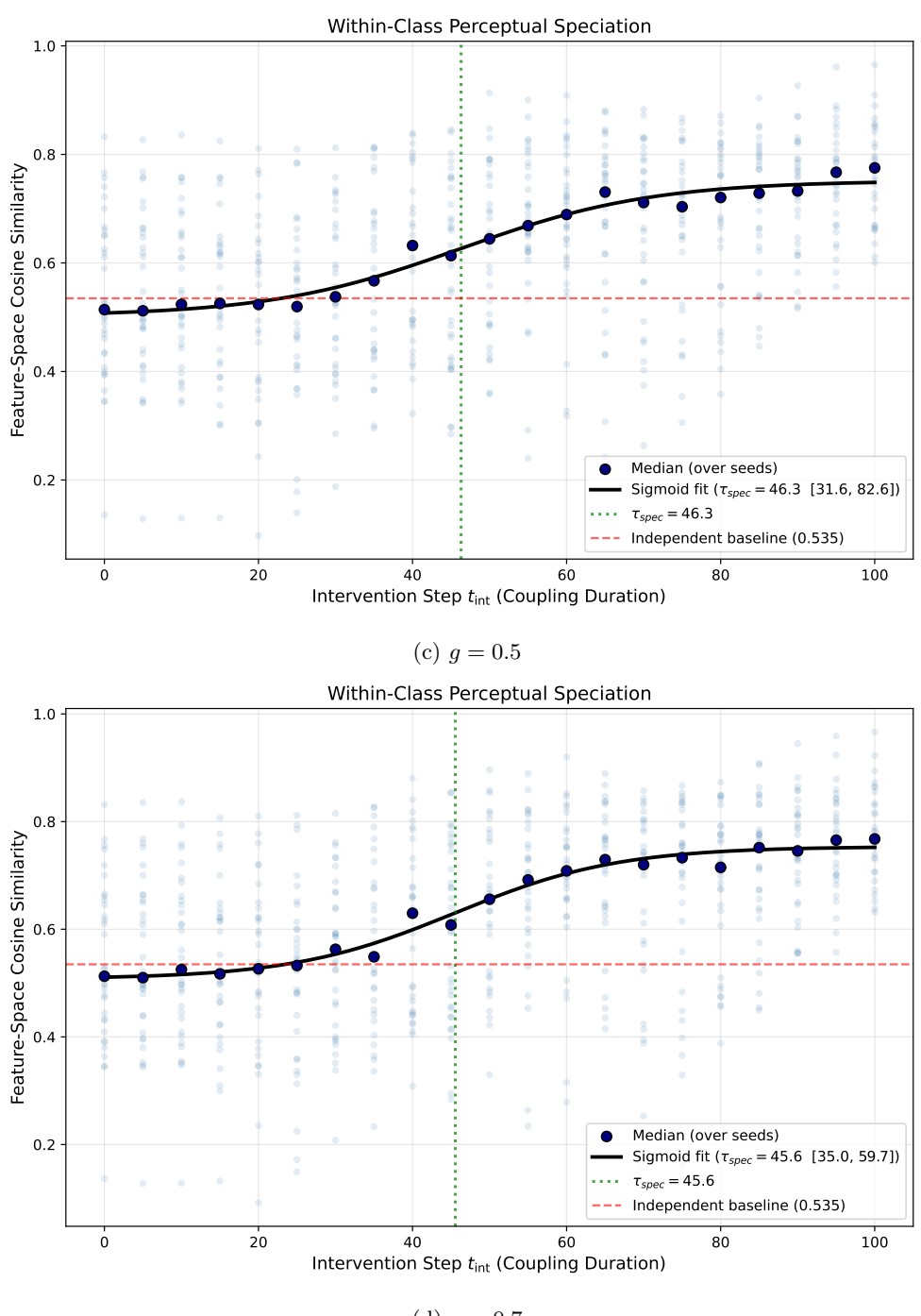

(c) $g = 0.5$

(d) $g = 0.7$

Figure 2: Each plot shows the median feature space cosine similarity Equation 70 between paired outputs as a function of intervention step $t_{\text{int}}$, together with the sigmoid fit used to extract $\tau_{spec}(g)$. With an increase of coupling strength $g$, the speciation time decreases.

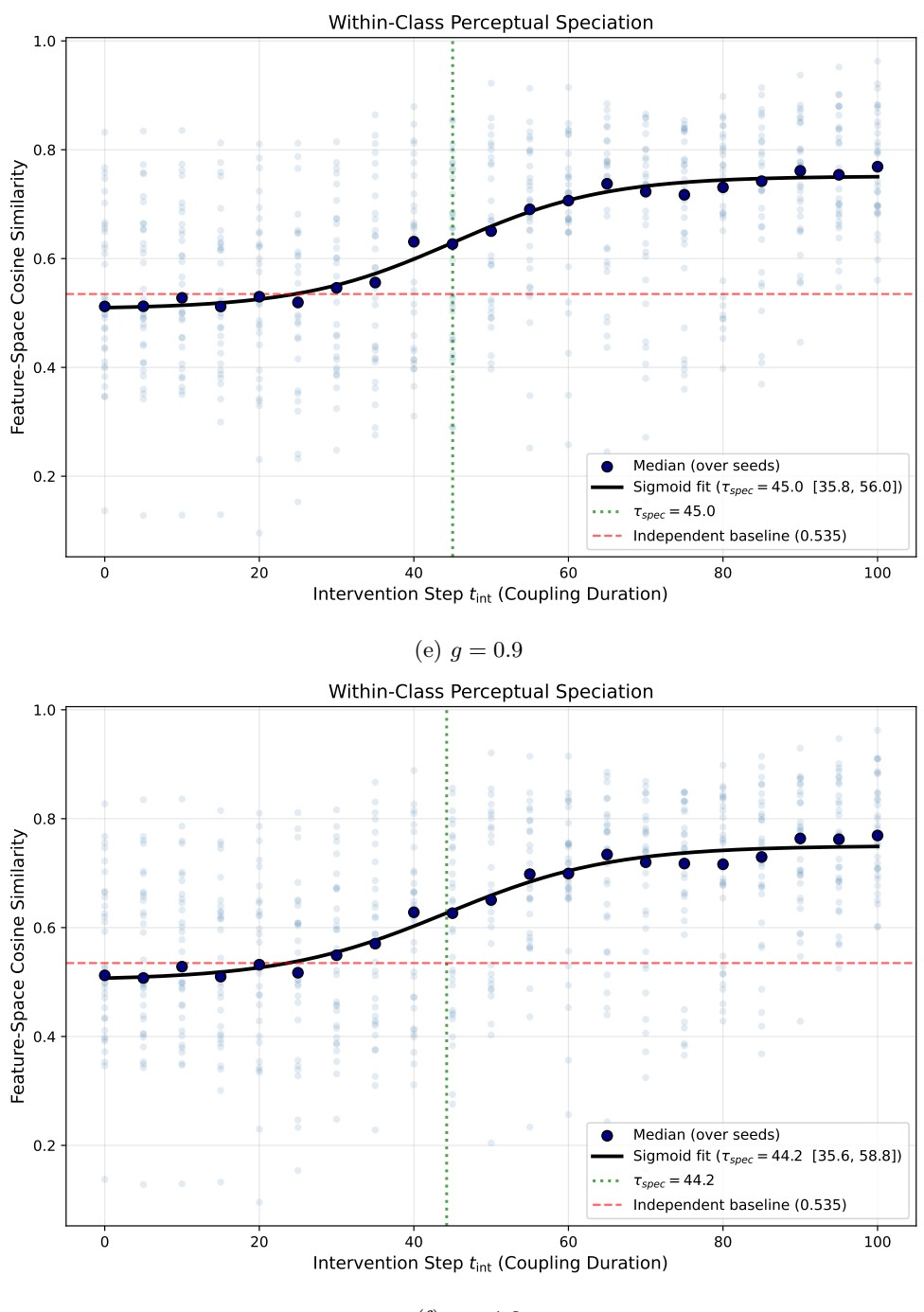

(e) $g = 0.9$

(f) $g = 1.0$

Figure 2: Each plot shows the median feature space cosine similarity Equation 70 between paired outputs as a function of intervention step $t_{\text{int}}$, together with the sigmoid fit used to extract $\tau_{spec}(g)$. With an increase of coupling strength $g$, the speciation time decreases.

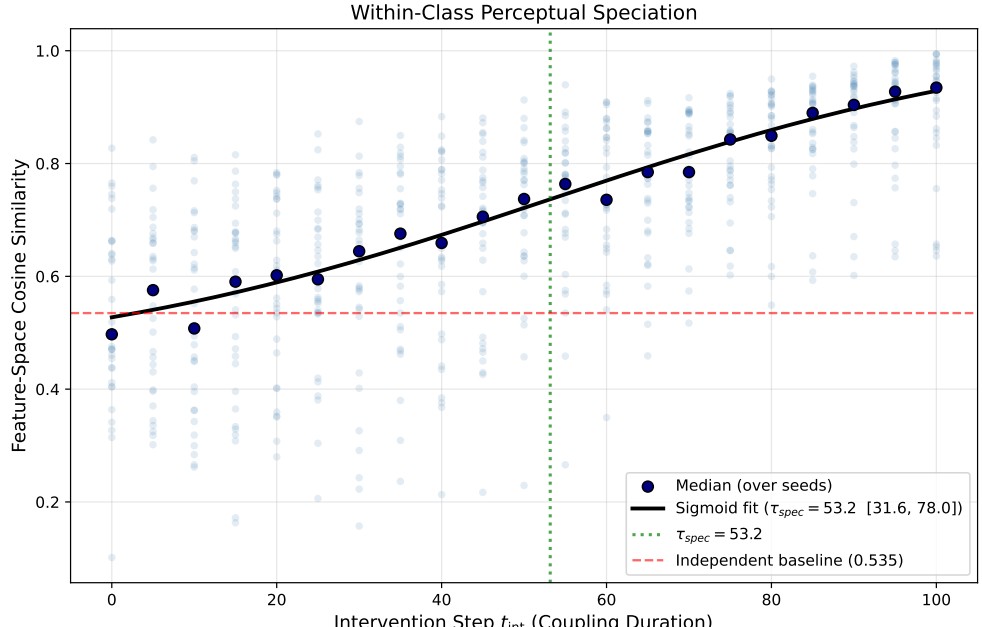

Figure 3: A separate analysis of the speciation time using the median feature space cosine similarity Equation 70 between paired outputs for the vanishing coupling strength $g = 0$. In this modified setup, the noise of reverse process for both replicas are shared until intervention time when it becomes independent.

in the trajectory and thereby reduces the temporal window over which the two replicas can diverge before committing to the same semantic basin.

We report only values of $g > 0$ since the experiment is about measuring the intervention time when $g$ is switched to 0. In the case of $g = 0$, we can plot a separate uncoupled control where instead two replicas share noise until intervention time when noise is switched to be independent. For such setup, we can again ask at what point these two trajectories finish at the same basin of the attraction and this is shown on figure 3. In this case, speciation time is earlier than for $g = 0.1$, but this is due to the different intervention type as explained above.

A second output of Protocol I is a measurement of a scale dependent output commitment. Here we use the coarse and fine commitment scores defined in Equation 71, Equation 72 respectively, and fit separate sigmoids to obtain the midpoint times of global and local output alignment. For readability, we denote these fitted midpoints by $\tau_{\mathrm{g}}(g)$ and $\tau_{\mathrm{l}}(g)$, corresponding to the blue and red curves in the decoupling plots. As in the previous case we sweep coupling strength $g \in \{0.1, 0.3, 0.5, 0.7, 0.9, 1.0\}$.

The measurements presented on figure 4a–figure 4f show that coarse output structure stabilizes substantially earlier than fine detail throughout the explored coupling range. Thus, the decoupling probe supports the existence of an output space gap. This gap is extremely robust; after the weak coupling regime, the gap stabilizes near $\Delta\tau \approx 39\text{--}41$ steps across the medium and strong coupling regime. Importantly, these scale dependent midpoint times should not be identified with the primary speciation time $\tau_{spec}(g)$. The decoupling probe measures when coarse and fine pixel space discrepancies become suppressed relative to an independent baseline. By construction, it is sensitive to the chosen low/high decomposition, the baseline, and residual variability. In contrast, $\tau_{spec}(g)$ is defined from cosine similarity in the feature space of a pretrained ResNet-50 encoder and is intended to measure semantic agreement between the final paired samples, i.e., whether the two trajectories have committed to the same basin of attraction. For this reason, we use the cosine similarity as the primary estimator of speciation time and treat the decoupling probe as a complementary diagnostic. The former is closer to the theoretical notion of branch commitment. The latter is still essential,

because it reveals the internal ordering of coarse versus fine output stabilization and thereby quantifies the observable synchronization gap in image space.

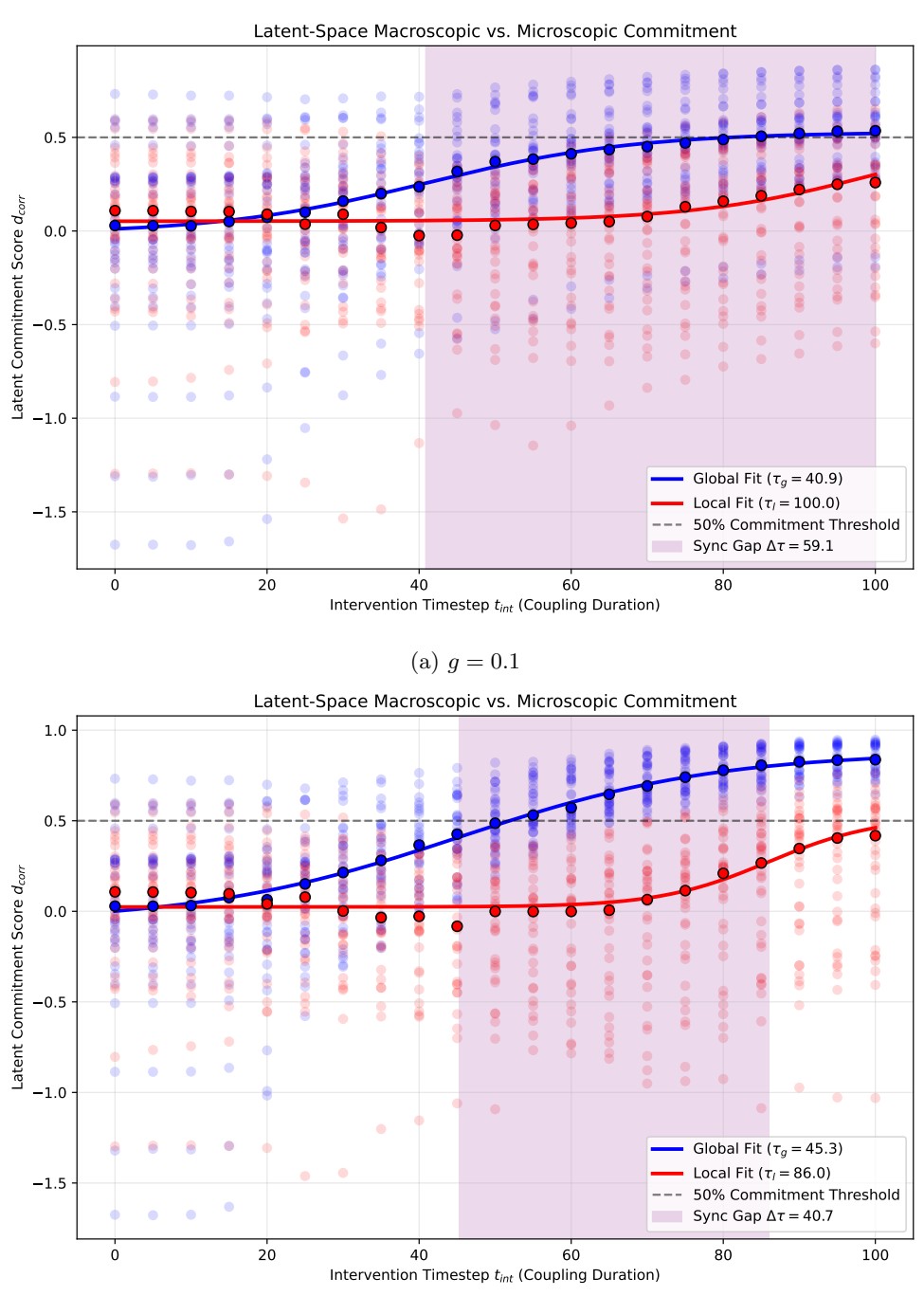

(a) $g = 0.1$

(b) $g = 0.3$

Figure 4: Each plot shows the global Equation 71 and local Equation 72 commitment scores as a function of intervention step $t_{\text{int}}$ (coupling duration), at coupling strength $g$. These show that coarse output structure stabilizes substantially earlier than fine detail throughout the explored coupling range.

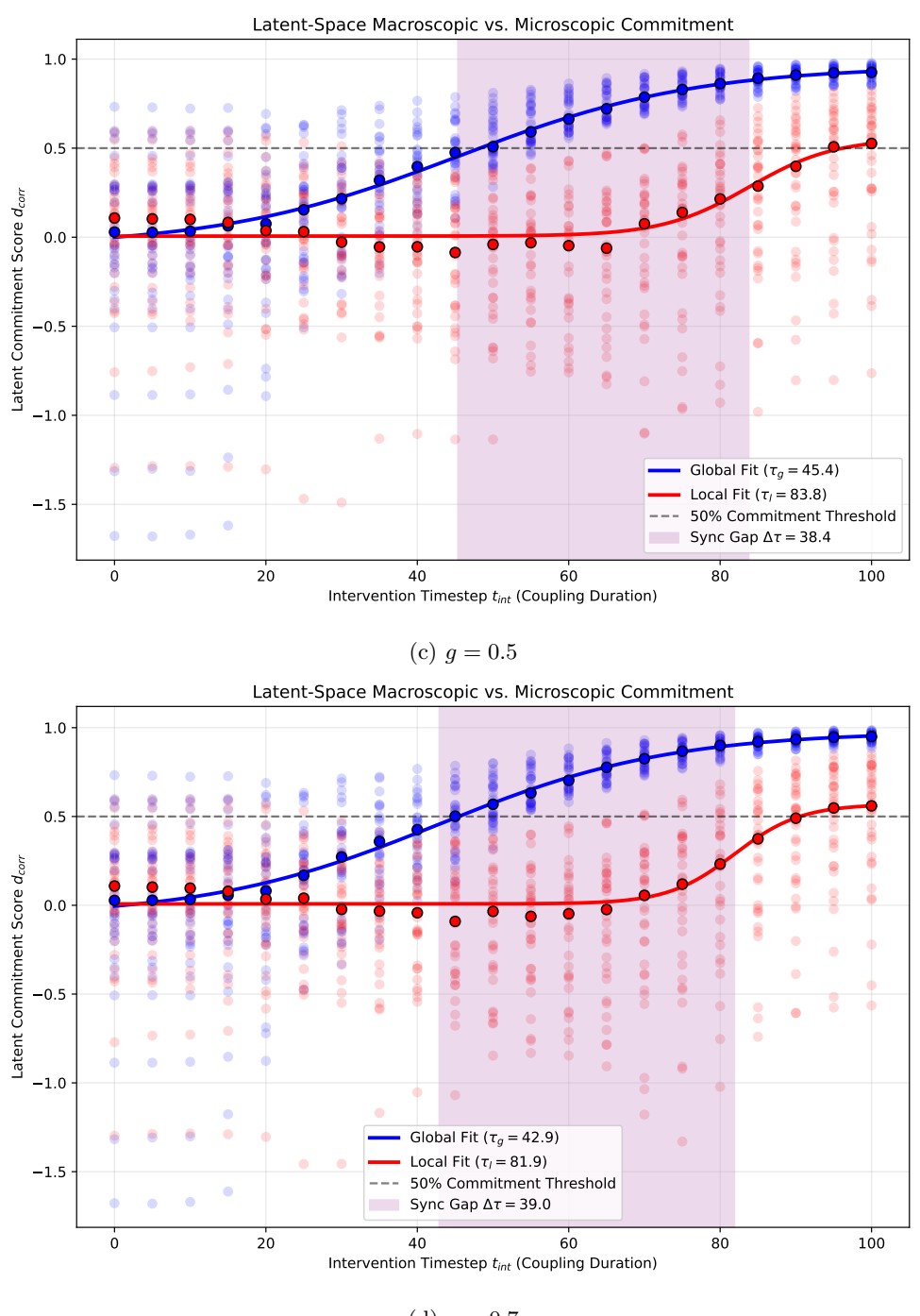

(c) $g = 0.5$

(d) $g = 0.7$

Figure 4: Each plot shows the global Equation 71 and local Equation 72 commitment scores as a function of intervention step $t_{\text{int}}$ (coupling duration), at coupling strength $g$. These show that coarse output structure stabilizes substantially earlier than fine detail throughout the explored coupling range.

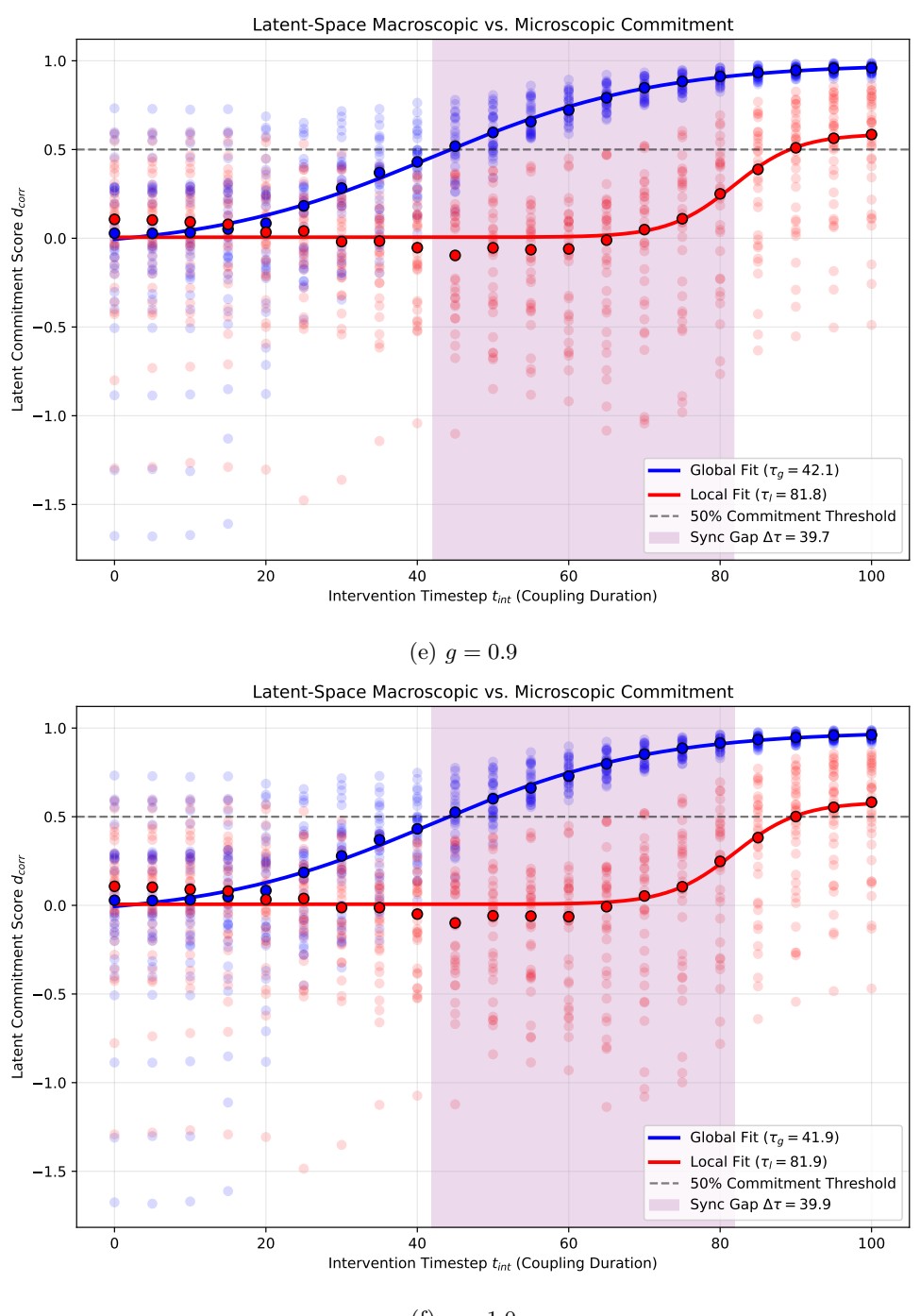

(e) $g = 0.9$

(f) $g = 1.0$

Figure 4: Each plot shows the global Equation 71 and local Equation 72 commitment scores as a function of intervention step $t_{\text{int}}$ (coupling duration), at coupling strength $g$. These show that coarse output structure stabilizes substantially earlier than fine detail throughout the explored coupling range.

### 4.2 Protocol II Results

We now discuss the central experimental result of this paper, given by Protocol II setup section 3.2, which provides the primary mechanistic test of the theory. For a particular coupling strength $g$, we evaluate the normalized fixed basis energies of leading and trailing internal difference modes Equation 78–Equation 79 at the speciation time identified in Protocol I experiment and sweep the capture layer across Transformer depth. In the case of DiT-XL/2, we sweep through all 28 layers. We use 100 reverse sampling steps with deterministic DDIM sampler $\eta = 0$ and we normalize replica attention as in Equation 19. We plot mode mean and $\pm\sigma$ spread from paired seeds. The central qualitative result is a strong coupling dependent suppression of the internal hidden state synchronization gap.

In the case $g = 0$, figure 1, the leading and trailing mode energies remain close through most of the network but separate sharply in the final blocks, where the trailing band retains substantially more energy than the leading band. Thus, the internal synchronization gap is strongly depth localized, it is weak in early and middle layers and becomes most evident only near the deepest blocks. The weak coupling case $g = 0.1$ exhibits the same qualitative pattern figure 5a, although the late layer split is somewhat reduced. Both $g = 0$ and $g = 0.1$ exhibit an inversion period, roughly between the 16th and 22nd layers, where the trailing mode is overtaken by the leading mode. In particular, for the case of $g = 0.0$, this period is characterized by large spread which indicates high dependence on random seeds. We interpret this behavior in the following way: because coupling $g$ is zero or small, some replica trajectories diverge wildly, causing the energy difference spike. However, later layers damp the chaotic internal variance and reestablish the proper leading/trailing hierarchy

In contrast, at $g = 0.3$, the two curves are already nearly indistinguishable across essentially the full depth of the network figure 5b, indicating that moderate coupling is sufficient to suppress most of the internal hierarchy predicted by the spatial routing dominant theory. At $g = 0.9$, figure 5c, the collapse is even more complete, and the leading and trailing mode energies are nearly superposed across depth, with only negligible residual differences.

These depth sweeps support the main theoretical claim of the paper. The effective spatial routing term in the linearized difference channel propagator is weighted by $\rho(g) = (1 - g)/(1 + g)$, Equation 28, so the hierarchy between leading and trailing modes should weaken as $g$ increases and should collapse in the strongly coupled regime. The data are consistent with this prediction and further sharpen it. Empirically, the collapse is not confined to the asymptotic regime $g \to 1$, but is already largely realized by moderate coupling.

A second important observation is that the internal gap is concentrated in late blocks. This depth localization is also consistent with the theoretical picture in which layer dependent modulation and residual composition cause the effective difference channel Jacobian to vary strongly across depth. In particular, the results suggest that the deepest transformer blocks are where the leading/trailing hierarchy becomes relevant in the uncoupled or weakly coupled regime, while strong coupling suppresses that hierarchy before it can fully emerge.

Taken together, the results support the following picture. Strong symmetric replica coupling sharply suppresses the internal, depth localized hierarchy between leading and trailing difference modes, which is in agreement with the spatial routing dominant theory. Nevertheless, after coupling is removed, fine output detail still requires a longer period of synchronized evolution than coarse structure in order to survive later independent stochastic refinement. The main novel message is therefore a two level synchronization phenomenon, coupling collapses the internal hidden state hierarchy, while a residual coarse to fine output commitment lag can persist in decoded image space.

### 4.3 Direct Measurement of the Attention Decomposition

Protocols I and II probe the consequences of the difference channel dynamics, the speciation times and the synchronization gap. To complete the picture and explain assumptions made we measure the ingredients of the theory directly on the pretrained network.

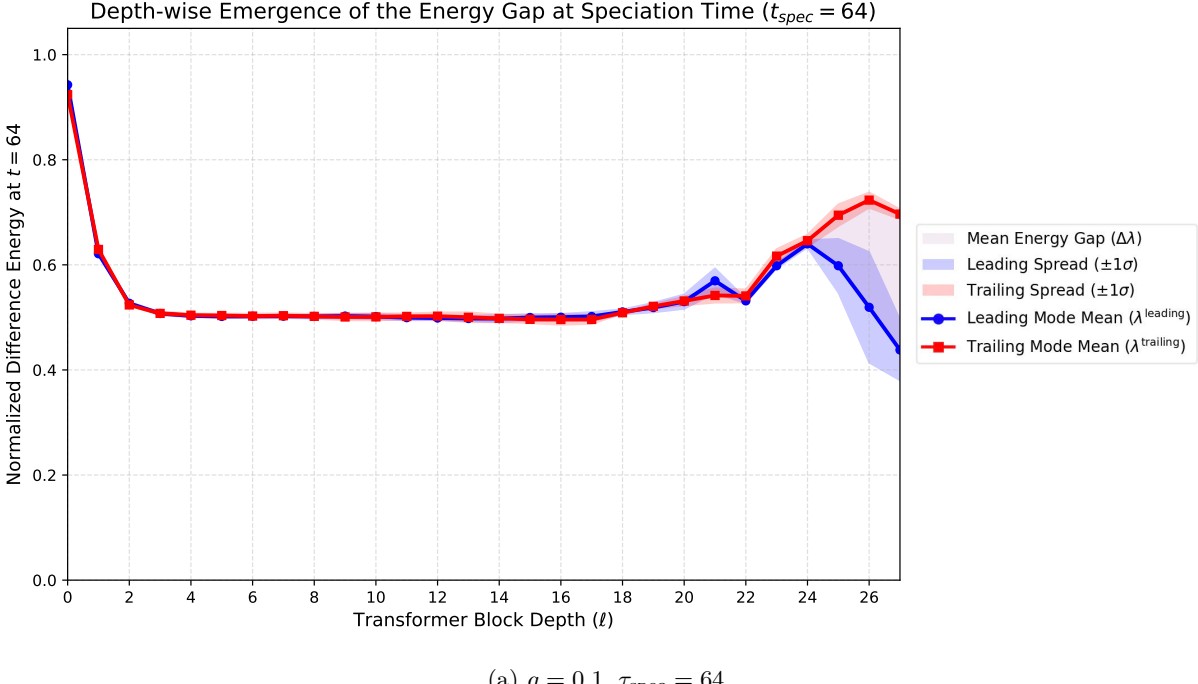

(a) $g = 0.1$, $\tau_{\mathrm{spec}} = 64$

Figure 5: For particular coupling strength $g$ and speciation time we sweep across all Transformer layers to evaluate the normalized fixed basis energies of leading and trailing internal difference modes Equation 78–Equation 79. These sweeps reveal that the gap is located at deep layers of the Transformer and as coupling strength $g$ increases the gap collapses.

We split the attention difference Equation 25 into its routing, modulation and residual contributions as

$$\Delta_{\mathrm{full}} = \Delta_{\mathrm{route}} + \Delta_{\mathrm{mod}} + \Delta_{\mathrm{resid}}, \tag{81}$$

and measure their fractions $r_{\mathrm{route}}, r_{\mathrm{mod}}, r_{\mathrm{resid}}$,

$$r_{\mathrm{route}} = \sqrt{\frac{\sum_{\mathrm{seeds}} ||\Delta_{\mathrm{route}}||^2}{\sum_{\mathrm{seeds}} ||\Delta_{\mathrm{full}}||^2}}, \quad r_{\mathrm{mod}} = \sqrt{\frac{\sum_{\mathrm{seeds}} ||\Delta_{\mathrm{mod}}||^2}{\sum_{\mathrm{seeds}} ||\Delta_{\mathrm{full}}||^2}}, \quad r_{\mathrm{resid}} = \sqrt{\frac{\sum_{\mathrm{seeds}} ||\Delta_{\mathrm{resid}}||^2}{\sum_{\mathrm{seeds}} ||\Delta_{\mathrm{full}}||^2}}, \tag{82}$$

over every $(\ell, s)$, we plot this on figure 6. At $g = 0$ in the early reverse phase routing dominates, $r_{\mathrm{route}} \approx 0.95$ versus $r_{\mathrm{mod}} \approx 0.27$ averaged over layers, and the routing panel vanishes identically at $g = 1$ where $\rho(1) = 0$, while modulation persists through $\xi(g) = 1/(1+g)$. The first order split is exact up to a genuinely nonlinear residual scaling as $\mathcal{O}(\sigma^3)$ as can be seen on figure 7, confirming the linear decomposition is accurate at the perturbation amplitudes used.

The same routing dominance and $\rho(g)$ collapse reproduce beyond the base configuration. On figure 8 we show these for DiT-XL/2-512, PixArt-$\Sigma$-256 and a second ImageNet class, the bare routing ratio at $g = 0$ is 0.43 (PixArt) versus 0.46 (DiT-XL/2), and routing collapses to zero at $g = 1$ along $\rho(g)$ in every case.

The residual gates $\alpha_\ell, \beta_\ell$ that scale the attention and MLP branches are $\mathcal{O}(1)$ and grow with depth (RMS 1.4–2.7), largest in the late blocks where the gap forms (figure 9). The local score gain, $\gamma_{s,\ell} \propto ||H_{\ell+1} - H_\ell|| / ||H_\ell||$, has median $\approx 0.17$ across $(\ell, s)$ and rises toward the late layers, confirming the monotonicity the predictions require. Because the gates are $\mathcal{O}(1)$ rather than infinitesimal, the propagator cross term is not negligible, it reaches $\approx 44\%$ of the one block update (median), peaking near 85% at mid layers (figure 10), which is why we retain $K_g$ in full Equation 32.

Finally, to test whether the late blocks are not merely where the gap is largest but to localize where it appears, we restrict the coupling Equation 19 to a set of blocks, leaving the rest at $g = 0$. Coupling only

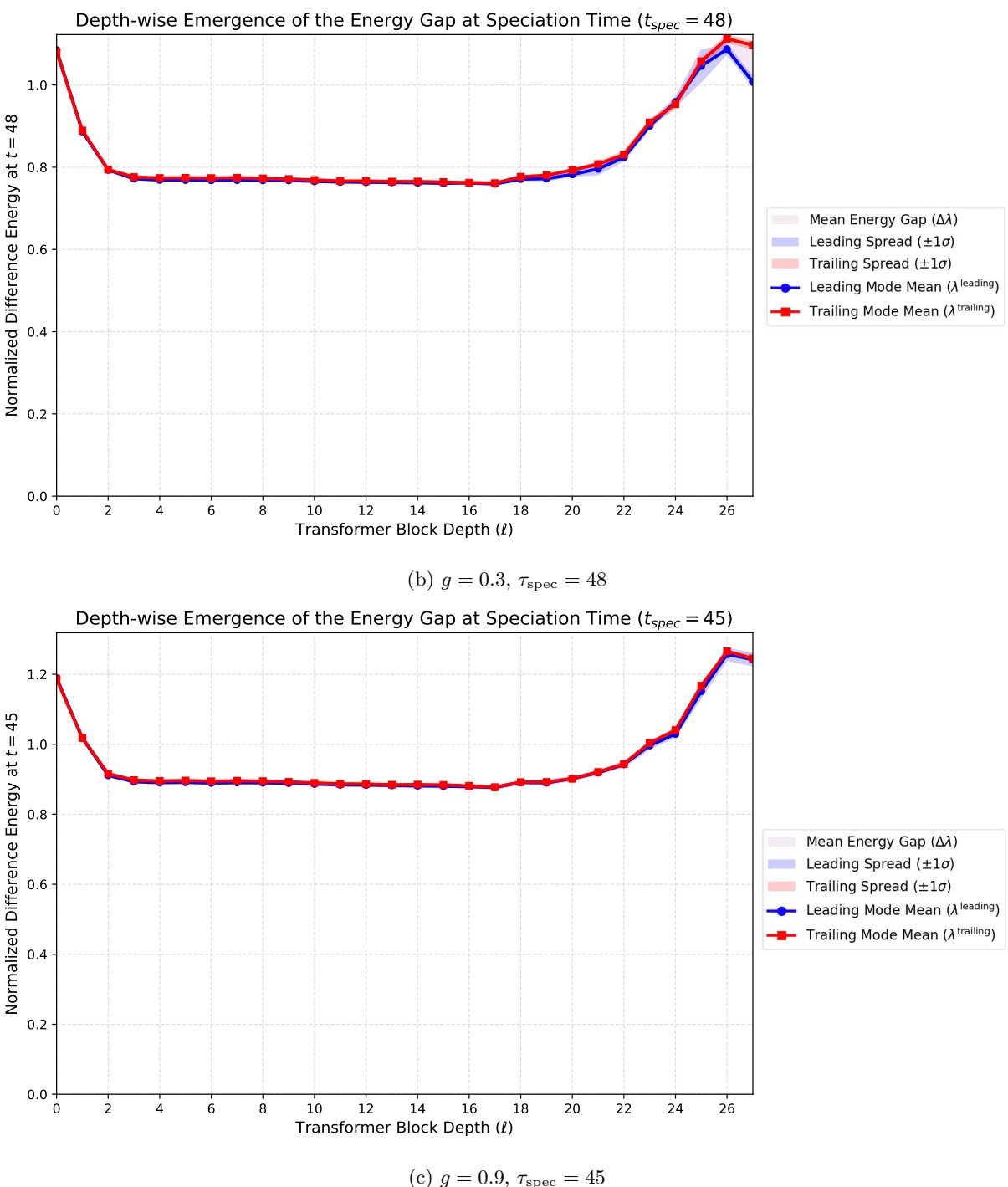

(b) $g = 0.3$, $\tau_{\mathrm{spec}} = 48$

(c) $g = 0.9$, $\tau_{\mathrm{spec}} = 45$

Figure 5: For particular coupling strength $g$ and speciation time, we sweep across all Transformer layers to evaluate the normalized fixed basis energies of leading and trailing internal difference modes Equation 78–Equation 79. These sweeps reveal that the gap is located at deep layers of the Transformer and as coupling strength $g$ increases the gap collapses.

the final 6 blocks reproduces the full coupling collapse, whereas the uncoupled gap is confined to those same blocks (figure 11). This localizes the mechanism, not just the observable, to the terminal blocks.

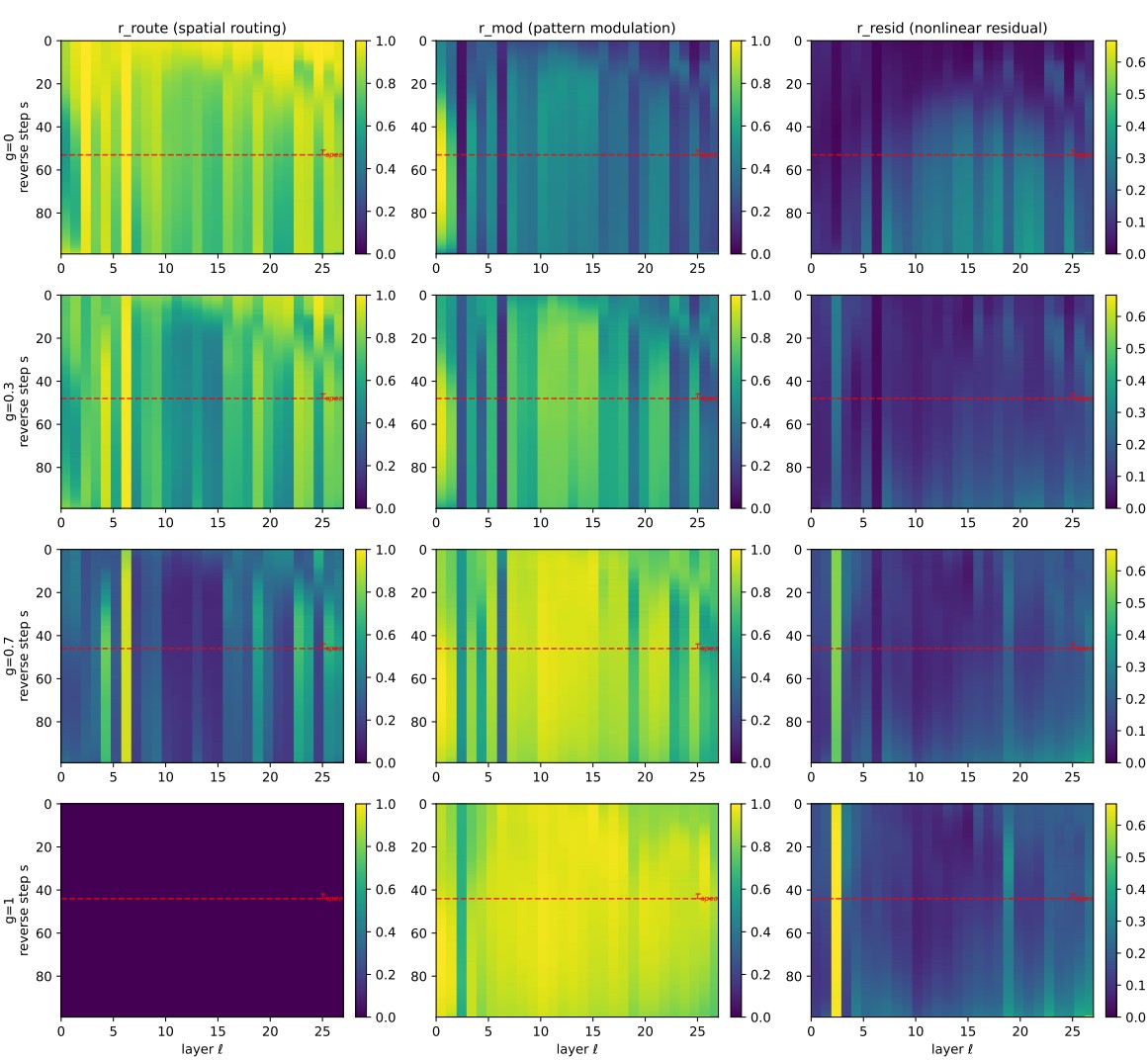

Figure 6: Equation 25 decomposition fractions $r_{\mathrm{route}}, r_{\mathrm{mod}}, r_{\mathrm{resid}}$ over $(\ell, s)$ for $g \in \{0, 0.3, 0.7, 1\}$. Routing dominates at $g{=}0$; the $g{=}1$ routing panel is identically zero ($\rho(1){=}0$).

## 5 Conclusions

We have presented a theoretical and empirical study of the synchronization gap in Diffusion Transformers. The main contribution is a controlled framework for analyzing how pretrained DiTs resolve generative ambiguity across spatial scales, connecting the statistical physics picture of coupled diffusion processes to the Transformer architecture.

On the theoretical side, we constructed an explicit architectural realization of symmetric replica coupling inside DiT self-attention Equation 19 and derived the linearized attention difference Equation 25, which decomposes the network's first order response to the replica difference mode into a spatial routing channel and a pattern modulation channel. We proved, using exact identities of the row stochastic softmax structure Appendix B, that the pattern modulation channel cannot see the constant token component of the common-mode value field while spatial routing preserves it, and we established by direct measurement in Appendix B that routing dominates for low frequency difference modes. To obtain a speciation criterion,

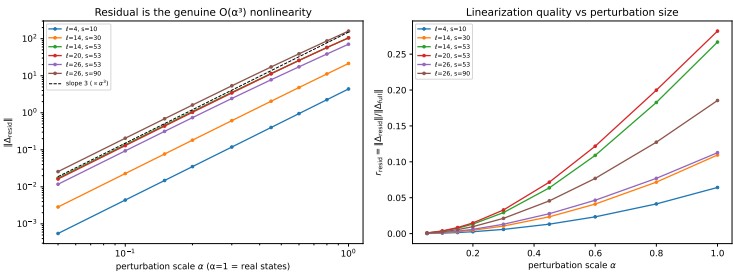

Figure 7: The residual of Equation 25 scales as $\mathcal{O}(\sigma^3)$ on real operating-point states: the linear routing/modulation split is exact and the residual is the genuine nonlinearity.

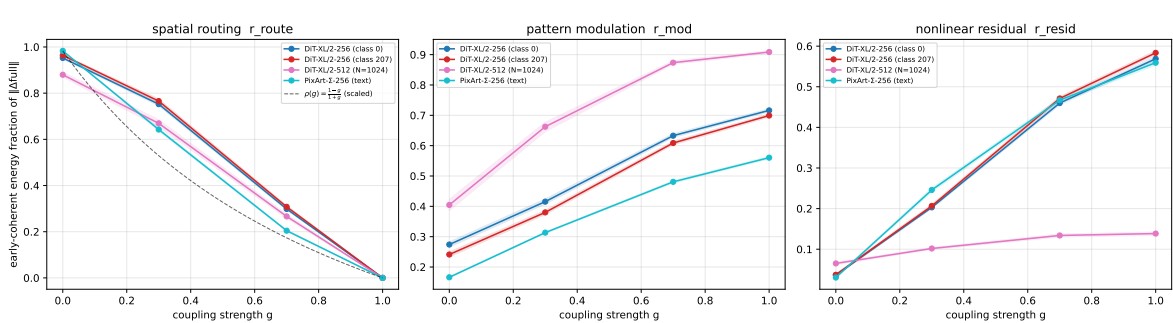

Figure 8: Routing/modulation/residual vs. $g$ in the early-denoise regime for DiT-XL/2-256 (ImageNet classes 0 and 207), DiT-XL/2-512, and PixArt-$\Sigma$-256. All models show routing dominance at $g=0$ collapsing to 0 at $g=1$ along $\rho(g)$.

we modeled the local difference mode distribution as a symmetric two component Gaussian mixture and projected the resulting fixed point equation onto empirical eigenmodes, yielding the scalar self consistency condition Equation 51 with a modewise speciation parameter that decomposes into an attention gated SNR Equation 53. Under the routing dominance assumption, the SNR difference between leading and trailing modes scales as $\mathcal{O}(\frac{1-g}{1+g})$, predicting the collapse of the internal synchronization gap at strong coupling.

On the empirical side, we tested these predictions on a pretrained DiT-XL/2 model using two complementary protocols. Protocol I section 3.1, based on a decoupling intervention with stochastic sampling, established that the speciation time decreases monotonically with coupling strength figure 2a–figure 2f and Table 1. The scale dependent output probe confirmed that global image features commit substantially earlier than local details across all tested coupling strengths, with an output space synchronization gap of $\Delta\tau \approx 39$–$41$ steps that is robust throughout the medium and strong coupling regime figure 4a–figure 4f. Protocol II section 3.2, based on a sweep across Transformer layers and measurement of internal difference mode energies at the speciation time, revealed three findings. First, the internal synchronization gap exists even at $g = 0$ and is concentrated in the final approximately six Transformer blocks figure 1–figure 5a. Second, moderate coupling ($g = 0.3$) is already sufficient to suppress most of the internal leading/trailing hierarchy figure 5b. Third, near strong coupling ($g = 0.9$), the leading and trailing mode energies are nearly superposed across the full network depth figure 5c.

Taken together, the results reveal a two level synchronization phenomenon. At the internal representation level, coupling collapses the depth localized hierarchy between leading and trailing difference channel modes, consistent with the suppression by the $\frac{1-g}{1+g}$ prefactor. At the output level, however, a residual global to local commitment lag persists in decoded image space even under strong coupling, indicating that the VAE decoder and the cumulative effect of many reverse steps introduce additional scale dependent processing that is not captured by the single block linearized theory.

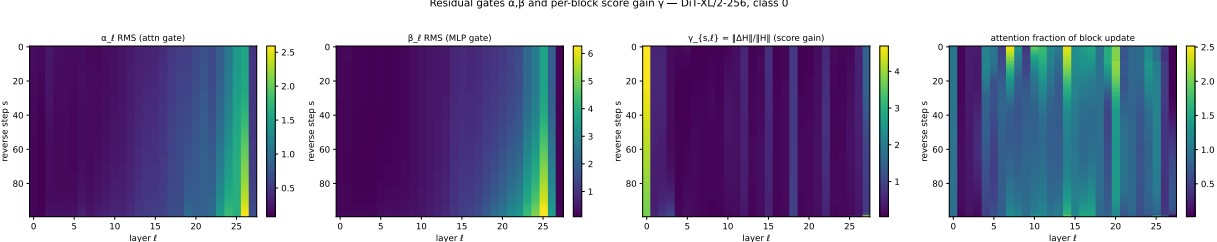

Figure 9: Residual gates $\alpha_\ell, \beta_\ell$ (RMS) and per-block score gain $\gamma_{s,\ell}$ over $(\ell, s)$. Gates are $\mathcal{O}(1)$ and grow with depth, largest at the late layers where the gap forms.

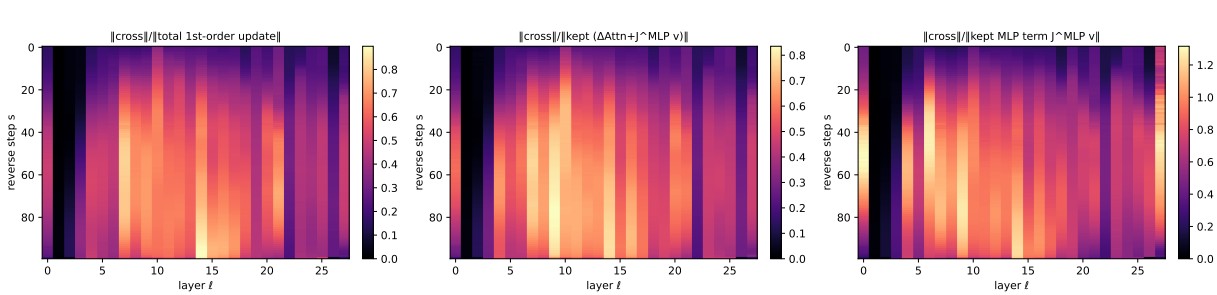

Figure 10: Magnitude of the $K_g$ cross term $J^{\mathrm{MLP}}(\rho R + \xi P)$ relative to the kept terms over $(\ell, s)$. It is $\approx 44\%$ of the block update (median), peaking $\sim 85\%$ at mid layers, confirming it is not negligible.

Several limitations of the present work should be noted. First, the local score gain $\gamma_{s,\ell}$, which controls the overall amplitude of the score contribution at each block Equation 36, is estimated here only through a residual block update proxy, which captures the monotonicity that the qualitative predictions require but not the absolute scale. A first principle calibration of $\gamma_{s,\ell}$ from the AdaLN modulation and the learned output projection scale, enabling quantitative prediction of absolute speciation times, remains for future work. Second, the mean field projection onto individual empirical modes neglects cross mode couplings that enter at $\mathcal{O}(\|v\|^3)$. Near the bifurcation point, where $|u_k|$ is small, these corrections are subleading, but when away from criticality, they may become relevant. Third, the diagonality assumption on $C_{s,\ell}$ Equation 44 in the empirical basis is a modeling approximation whose validity depends on the alignment between the mixture covariance and the initial difference covariance. Fourth, beyond DiT-XL/2-256 we reproduce routing/modulation dominance and the collapse of the gap on DiT-XL/2-512, PixArt-$\Sigma$-256, and a second ImageNet class. Broader verification across modalities, e.g. video and 3D-molecular DiTs, would further strengthen generality.

The framework developed here opens several directions for future investigation. The observation that the synchronization gap is generated predominantly in the terminal Transformer blocks suggests that targeted interventions at specific layers and reverse time steps could be used to selectively modify the commitment structure of the generative process, with potential applications to controlled generation and concept editing. More broadly, the layerwise propagator $K_g$ Equation 32 and the modewise SNR formula Equation 53 provide a quantitative language for analyzing how spatial information is routed through the Transformer depth during generation, connecting the macroscopic phenomenology of phase transitions in diffusion models to the microscopic mechanics of self attention. Finally, the explicit connection between the attention gating prefactor and the thermodynamic cost of synchronization, suggested by the dissipative nature of the coupling term, points toward a stochastic thermodynamic (Limmer, 2024) characterization of the generative process that we leave for future work.

Furthermore, our mechanistic characterization of the synchronization gap suggests a principled interpretation of recent training free acceleration methods based on temporal feature forecasting and feature reuse (Liu, Zou,

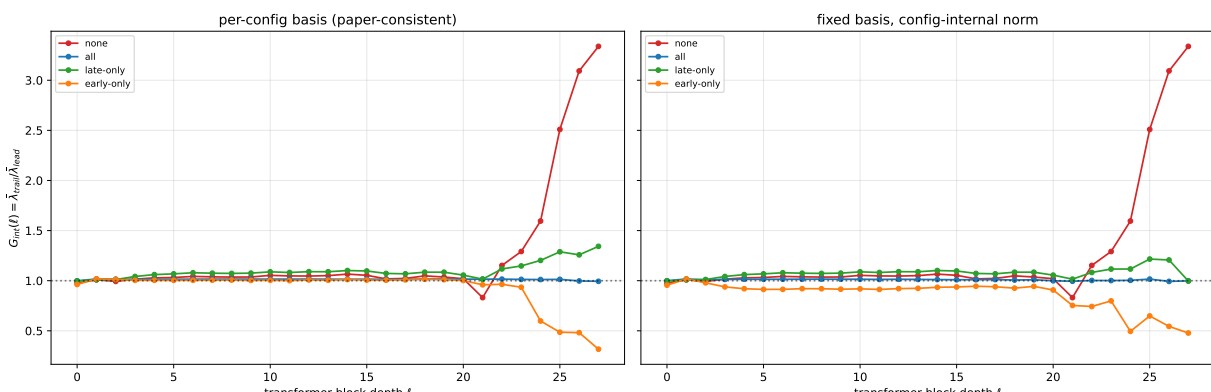

Figure 11: Layer restricted coupling. The gap (red, $g$=0) emerges only in the final ~6 blocks. Coupling only the late blocks (green) reproduces the full coupling collapse (blue), coupling only early blocks (orange) also suppresses it. Per configuration and fixed basis results agree.

Lyu, Chen, and Zhang, 2025; Han, Shi, Li, Ye, Guo, and Ermon, 2026). In particular, our results indicate that commitment of trailing modes is delayed relative to leading modes and that the corresponding internal hierarchy is concentrated in the deepest Transformer blocks. This provides a structural explanation for why temporal approximation can preserve global semantics while degrading local detail, errors introduced during the late stage evolution of trailing modes, especially in terminal blocks, are expected to have a disproportionate effect on fine image fidelity. From this perspective, feature caching strategies should be stage and depth aware, with more reuse in early regimes and exact evaluation retained during late stage refinement and in the final blocks where fine detail commitment is resolved.

## A  Detailed Derivation of Equation 25

In this appendix, we suppress the layer index $\ell$. The attention output for replica $A$ is:

$$\text{Attn}_g^A = \frac{1}{1+g}\left[A_{AA}\, V_A + g\, A_{AB}\, V_B\right]. \tag{83}$$

Substituting the first-order expansions:

$$A_{AA}\, V_A = (A_0 + \delta A^{(+)})(V_0 + \delta V) = A_0 V_0 + A_0\, \delta V + \delta A^{(+)} V_0 + \mathcal{O}(\|v\|^2), \tag{84}$$

$$A_{AB}\, V_B = (A_0 + \delta A^{(-)})(V_0 - \delta V) = A_0 V_0 - A_0\, \delta V + \delta A^{(-)} V_0 + \mathcal{O}(\|v\|^2). \tag{85}$$

Summing these terms yields:

$$\text{Attn}_g^A = \frac{1}{1+g}\left[(1+g)\, A_0 V_0 + (1-g)\, A_0\, \delta V + \left(\delta A^{(+)} + g\, \delta A^{(-)}\right) V_0\right] + \mathcal{O}(\|v\|^2). \tag{86}$$

By the exact exchange symmetry of the system, $\text{Attn}_g^B$ is obtained by mapping $v \to -v$, which flips the signs of the linear perturbations ($\delta V \to -\delta V$ and $\delta A^{(\pm)} \to -\delta A^{(\pm)}$). Taking the difference $\text{Attn}_g^A - \text{Attn}_g^B$ cancels the base term $A_0 V_0$ and doubles the linear terms, yielding the desired result Equation 25.

## B  Routing vs. Pattern Modulation in the Attention Difference

Below we provide a justification for one of the key structural assumptions that we use to analyze the synchronization gap collapse mechanism. In section 2.6 we stated that for low frequency difference channel $v$ modes the spatial routing term in the attention difference equation 25 dominates the pattern modulation term. The

argument has two parts, separating what the architecture fixes from what the trained weights determine. The first part discusses a pair of exact, non perturbative identities that follow solely from row stochasticity of softmax. They fix the structure of the two channels, in particular their behavior on the constant token subspace, independently of any learned quantity. The second addresses their relative magnitude, which is a property of the learned attention map $A_0$ rather than of the architecture and is therefore established by a direct measurement. Routing dominates because the antisymmetric weight perturbation is small and $A_0$ acts as a low pass filter on the modes that carry the difference energy, the same low pass action that yields the frequency selectivity inequality $\chi_{k_{\mathrm{hi}}} > \chi_{k_{\mathrm{lo}}}$ Equation 61 and hence the commitment hierarchy. As in the theoretical framework section, we view token space matrices such as $V_0$ and $\delta V$ as elements of $\mathbb{R}^{N \times d_{\mathrm{model}}}$.

Let $\mathbf{1} \in \mathbb{R}^N$ denote the token vector of all ones and define the orthogonal projectors

$$P_0 := \frac{1}{N} \mathbf{1}\mathbf{1}^\top, \qquad P_\perp := I - P_0. \tag{87}$$

Thus $P_0$ projects onto the subspace of constant tokens and $P_\perp$ onto the mean free tokens subspace. For any token space matrix $X \in \mathbb{R}^{N \times d_{\mathrm{model}}}$, we have the decomposition

$$X = P_0 X + P_\perp X. \tag{88}$$

Let $A, A' \in \mathbb{R}^{N \times N}$ be row stochastic, i.e. $A \geq 0$ entrywise and $A\mathbf{1} = \mathbf{1}$. Then we have $A P_0 = P_0$, and $(A - A')\,P_0 = 0$, equivalently $A - A' = (A - A')\,P_\perp$. Using that $AP_0 = \frac{1}{N}(A\mathbf{1})\mathbf{1}^\top = \frac{1}{N}\mathbf{1}\mathbf{1}^\top = P_0$ and $(A - A')\mathbf{1} = \mathbf{1} - \mathbf{1} = 0$, hence $(A - A')P_0 = \frac{1}{N}\big((A - A')\mathbf{1}\big)\mathbf{1}^\top = 0$, where the complementary form follows from $P_0 + P_\perp = I$. Each block softmax $A_{AA}, A_{BB}, A_{AB}, A_{BA}$ and the symmetric point map $A_0$ is row stochastic by construction Equation 18.

Now $\delta A$ is the first order perturbation Equations 23- 24 induced by a logit perturbation $\delta S$. Since blockwise softmax normalizes each of them separately, let us denote both as $\delta A$. From the construction above, we find

$$\delta A V_0 = \delta A P_\perp V_0. \tag{89}$$

Now for the spatial routing term we obtain

$$A_0 \delta V = P_0 \delta V + A_0 P_\perp \delta V. \tag{90}$$

Equation 89 and Equation 90 provide the structural core. Pattern modulation cannot see the constant token component of the common mode value field at all, while routing transmits unchanged the constant token component of the difference field and only filters its mean free part. The two channels are therefore complementary on the subspace of constant token and independent of any coherency or attention width assumption. We confirm both relations as operator identities on the pretrained network and they match up to numerical precision error as shown on figure 13.

## B.1   Relative size of the two channels

The identities Equations 89 and 90 fix the structure of the two channels but not their relative magnitude, and here two regimes must be separated. On the constant token subspace the magnitudes are fixed by the architecture alone, modulation vanishes identically and routing has exactly unit gain, for any row stochastic $A_0$. Off this subspace, the relative magnitude is governed by the learned attention map $A_0$ and by the spectral content of the difference field. It is a property of the trained weights, not a consequence of the architecture, and so cannot be settled a priori. We determine it by direct measurement on the pretrained model which, for a learned operator.

We can answer the magnitude question in following way. Using the modulation identity $\delta A = \delta A\, P_\perp$ together with submultiplicativity,

$$\frac{\|\delta A\, V_0\|}{\|A_0\, \delta V\|} \leq \|\delta A\|\, \frac{\|P_\perp V_0\|}{\|A_0\, \delta V\|}, \tag{91}$$

which is exact and factorizes routing dominance into two measurable causes: a small antisymmetric weight perturbation $\delta A$, and a large routing magnitude $\|A_0\delta V\|$. We read the ratio off the network directly. In the early denoising time ($g = 0$),

$$\frac{\|\delta A\, V_0\|}{\|A_0\,\delta V\|} \approx 0.46, \qquad \left\langle \frac{\|\delta A\, V_0\|^2}{\|A_0\,\delta V\|^2} \right\rangle_{(\ell,s)} \approx 0.21, \tag{92}$$

so modulation is about half of routing in operator norm and below a third in energy, routing dominates.

Both factors in Equation 91 have a mechanistic explanation. The antisymmetric perturbation $\delta A$ is small because the two replicas share the symmetric point query/key geometry, so $A_{AA} \approx A_{BB}$ and $A_{AB} \approx A_{BA}$, the inter-replica difference is carried predominantly by routing the differing values $\delta V$ through the shared map $A_0$, not by a change in the map. The routing magnitude stays large for a spectral reason. The mean free part of the difference field is not small in amplitude ($\|P_\perp \delta V\|/\|P_0 \delta V\| \approx 4.4$), but it is spectrally concentrated at low frequency, where $A_0$ retains it with near unit gain. Expanding $\delta V = \sum_k a_k r_k$ in the empirical difference modes ordered by increasing spatial frequency,

$$\|A_0\,\delta V\|^2 \approx \sum_k \chi_k^2\, a_k^2, \tag{93}$$

remains large because the high $a_k$ (low frequency) modes are precisely those with $\chi_k$ near its maximum. The decomposition is therefore in terms of eigenmode and frequency. Low frequency concentration is what the coherent low frequency regime means, and the two are distinct. A small amplitude fluctuation can be high frequency, and a large amplitude one can be low frequency.

The same mechanism supplies Equation 61, and the frequency selectivity resides in the row stochastic spatial map $A_0$ itself. The scalar routing gain $\chi_k = r_k^\top R\, r_k$ with $R = 2A_0 J_V$ (Equations 26 and 47) also folds in the value Jacobian $J_V$ (the channel maps $W_v, W_o$), which mixes channels and is sign indefinite, so $\chi_k$ is not by itself a clean frequency diagnostic. We therefore measure the spatial low pass gain of $A_0$ directly, its retention of each empirical difference mode $r_k$ figure 12. Retention falls monotonically with mode spatial frequency (Pearson correlation $-0.95$ at $s = 0$), with unit constant field retention and zero checkerboard retention. Leading modes retain more than trailing in most of layers, and are independently confirmed to be lower spatial frequency (radial centroid 0.29 vs. 0.37). The exact identities Equations 89 and 90 pin the zero frequency endpoint of this curve (routing gain unity, modulation gain zero), the measured retention fills in the interior. Because $\chi_k$ inherits this spatial profile through $A_0$, the coarse graining inequality $\chi_{k_{\mathrm{hi}}} > \chi_{k_{\mathrm{lo}}}$ (Equation 61) holds as an empirical property of the pretrained map rather than a posited one.

The low pass property is the common origin of both routing dominance and the commitment hierarchy of §2.7. Substituting $\chi_{k_{\mathrm{hi}}} > \chi_{k_{\mathrm{lo}}}$ into the routing dominant SNR (Equation 63),

$$\mathrm{SNR}_{v,k}(s,\ell;g) \approx \frac{m_k^2/c_k}{\left( -\lambda^{\mathrm{MLP}} - \rho(g)\,\chi_k\right)c_k + \gamma_{s,\ell}}, \qquad \rho(g) = \frac{1-g}{1+g} > 0 \;\; (g < 1), \tag{94}$$

the denominator is strictly decreasing in $\chi_k$, so $\partial\,\mathrm{SNR}_{v,k}/\partial\chi_k > 0$. At comparable $(m_k, c_k)$ this gives $\mathrm{SNR}_{v,k_{\mathrm{hi}}} > \mathrm{SNR}_{v,k_{\mathrm{lo}}}$, hence $\kappa_{v,k_{\mathrm{hi}}} > \kappa_{v,k_{\mathrm{lo}}}$ via $\kappa_{v,k} = \gamma_{s,\ell}\mathrm{SNR}_{v,k}$ (Equation 52), and, through the speciation condition $\kappa_{v,k}(s_{\mathrm{spec}}^{(k)}) = 1$ (Equation 51), $s_{\mathrm{spec}}^{(k_{\mathrm{hi}})} < s_{\mathrm{spec}}^{(k_{\mathrm{lo}})}$ (Equation 64). Low frequency modes commit earlier because $A_0$ routes them more strongly.

In the first version of this paper we tried to check Equation 91 by bounding $\|\delta A\|$ through the per row effective attention width $N_{\mathrm{eff}}^{(i)} = \left( \sum_j A_{0,ij}^2 \right)^{-1}$ and the coherency ratios $\varepsilon_{V_0} = \|P_\perp V_0\|/\|P_0 V_0\|$, $\varepsilon_{\delta V} = \|P_\perp \delta V\|/\|P_0 \delta V\|$. On the pretrained network these are $N_{\mathrm{eff}}^{\mathrm{min}} \approx 1$ and $\varepsilon_{V_0} \approx 3$, $\varepsilon_{\delta V} \approx 4.4$ (figure 13), so the resulting the worst case prefactor $2\sqrt{N}/\sqrt{N_{\mathrm{eff}}^{\mathrm{min}}} \approx 30$ is amplifying rather than suppressive, the worst case bound cannot establish dominance here. We therefore establish dominance by the direct measurement Equation 92.

## References

Beatrice Achilli, Marco Benedetti, Giulio Biroli, and Marc Mézard. Theory of speciation transitions in diffusion models with general class structure. *arXiv preprint arXiv:2602.04404*, 2026.

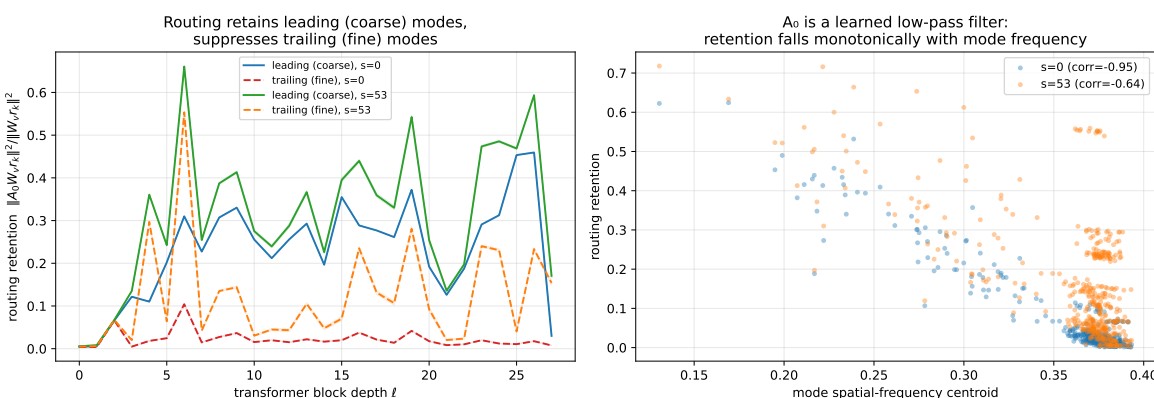

Figure 12: Spatial routing is a learned low-pass operator. Left: retention of leading modes exceeds trailing at all depths. Right: retention falls monotonically with mode spatial frequency (correlation $-0.95$).

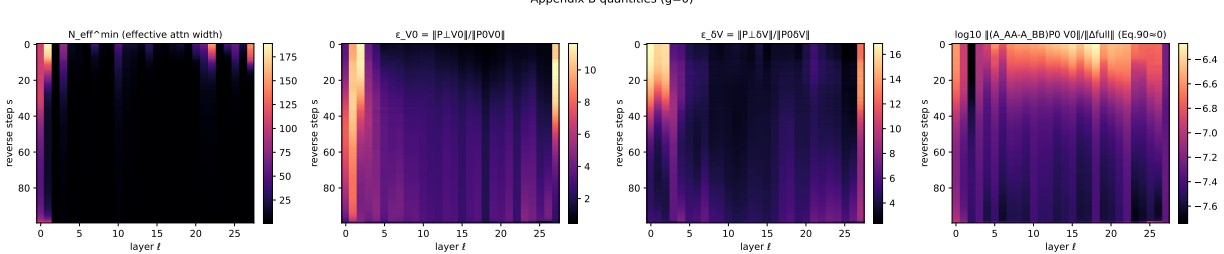

Figure 13: Measured $N_{\text{eff}}^{\min}$, coherency ratios $\varepsilon_{V_0}, \varepsilon_{\delta V}$, and the exact-identity residual of Equation 89 ($\sim 10^{-7}$, `float32`). $N_{\text{eff}}^{\min} \approx 1$ and $\varepsilon \approx 3$ make a worst case routing dominance bound amplifying rather than suppressive, which is why we establish dominance by direct measurement (Equation 92).

Amira Alakhdar, Barnabas Póczos, and Newell R. Washburn. Diffusion models in De Novo drug design. *Journal of Chemical Information and Modeling*, 64(19):7238–7256, 2024.

Emil Albrychiewicz, Andrés Franco Valiente, and Li-Ching Chen. Dynamical regimes of multimodal diffusion models. *arXiv preprint arXiv:2602.04780*, 2026.

Giulio Biroli, Tony Bonnaire, Valentin De Bortoli, and Marc Mézard. Dynamical regimes of diffusion models. *Nature Communications*, 15(1):9957, 2024.

Tim Brooks, Bill Peebles, Connor Holmes, Will DePue, Yufei Guo, Li Jing, David Schnurr, Joe Taylor, Troy Luhman, Eric Luhman, Clarence Ng, Ricky Wang, and Aditya Ramesh. Video generation models as world simulators. 2024. URL https://openai.com/research/video-generation-models-as-world-simulators.

Junsong Chen, Chongjian Ge, Enze Xie, Yue Wu, Lewei Yao, Xiaozhe Ren, Zhongdao Wang, Ping Luo, Huchuan Lu, and Zhenguo Li. Pixart-$\sigma$: Weak-to-strong training of diffusion transformer for 4k text-to-image generation, 2024.

Jia Deng, Wei Dong, Richard Socher, Li-Jia Li, Kai Li, and Li Fei-Fei. Imagenet: A large-scale hierarchical image database. In *2009 IEEE conference on computer vision and pattern recognition*, pp. 248–255. IEEE, 2009.

Mohammad Ennab and Hamid Mcheick. Enhancing interpretability and accuracy of ai models in healthcare: A comprehensive review on challenges and future directions. *Frontiers in Robotics and AI*, 11:1444763, 2024.

Polat Goktas and Andrzej Grzybowski. Shaping the future of healthcare: Ethical clinical challenges and pathways to trustworthy ai. *Journal of Clinical Medicine*, 14(5):1605, 2025.

Jiaqi Han, Juntong Shi, Puheng Li, Haotian Ye, Qiushan Guo, and Stefano Ermon. Adaptive spectral feature forecasting for diffusion sampling acceleration. *arXiv preprint arXiv:2603.01623*, 2026.

Florian Handke, Dejan Stančević, Felix Koulischer, Thomas Demeester, and Luca Ambrogioni. The entropic signature of class speciation in diffusion models. *arXiv preprint arXiv:2602.09651*, 2026.

Kaiming He, Xiangyu Zhang, Shaoqing Ren, and Jian Sun. Deep residual learning for image recognition. In *Proceedings of the IEEE Conference on Computer Vision and Pattern Recognition*, pp. 770–778, 2016.

Jonathan Ho, Ajay Jain, and Pieter Abbeel. Denoising diffusion probabilistic models. *Advances in Neural Information Processing Systems*, 33:6840–6851, 2020.

Jonathan Ho, William Chan, Chitwan Saharia, Jay Whang, Ruiqi Gao, Alexey Gritsenko, Diederik P. Kingma, Ben Poole, Mohammad Norouzi, David J. Fleet, and Tim Salimans. Imagen video: High definition video generation with diffusion models. *arXiv preprint arXiv:2210.02303*, 2022.

Mason Kamb and Surya Ganguli. An analytic theory of creativity in convolutional diffusion models. *arXiv preprint arXiv:2412.20292*, 2024.

Amirhossein Kazerouni, Ehsan Khodapanah Aghdam, Moein Heidari, Reza Azad, Mohsen Fayyaz, Ilker Hacihaliloglu, and Dorit Merhof. Diffusion models in medical imaging: A comprehensive survey. *Medical Image Analysis*, 88:102846, 2023.

Charles Kittel and Paul McEuen. *Introduction to Solid State Physics*. John Wiley & Sons, 2018.

Chieh-Hsin Lai, Yang Song, Dongjun Kim, Yuki Mitsufuji, and Stefano Ermon. The principles of diffusion models. *arXiv preprint arXiv:2510.21890*, 2025.

David T. Limmer. *Statistical Mechanics and Stochastic Thermodynamics: A Textbook on Modern Approaches in and out of Equilibrium*. Oxford University Press, 2024.

Pantelis Linardatos, Vasilis Papastefanopoulos, and Sotiris Kotsiantis. Explainable ai: A review of machine learning interpretability methods. *Entropy*, 23(1):18, 2020.

Jiacheng Liu, Chang Zou, Yuanhuiyi Lyu, Junjie Chen, and Linfeng Zhang. From reusing to forecasting: Accelerating diffusion models with taylorseers. In *Proceedings of the IEEE/CVF International Conference on Computer Vision*, pp. 15853–15863, 2025.

Haiqi Lu and Ying Tang. Steering dynamical regimes of diffusion models by breaking detailed balance. *arXiv preprint arXiv:2602.15914*, 2026.

Yuchen Ma, Valentyn Melnychuk, Jonas Schweisthal, and Stefan Feuerriegel. Diffpo: A causal diffusion model for learning distributions of potential outcomes. In *Advances in Neural Information Processing Systems*, volume 37, 2024.

Kanta Masuki and Yuto Ashida. Generative diffusion model with inverse renormalization group flows. *arXiv preprint arXiv:2501.09064*, 2025.

Marc Mezard and Andrea Montanari. *Information, Physics, and Computation*. Oxford University Press, 2009.

Marc Mézard, Giorgio Parisi, and Miguel Angel Virasoro. *Spin Glass Theory and Beyond: An Introduction to the Replica Method and Its Applications*. World Scientific Publishing Company, 1987.

William Peebles and Saining Xie. Scalable diffusion models with transformers. In *Proceedings of the IEEE/CVF International Conference on Computer Vision*, pp. 4195–4205, 2023.

Gabriel Raya and Luca Ambrogioni. Spontaneous symmetry breaking in generative diffusion models. *Advances in Neural Information Processing Systems*, 36:66377–66389, 2023.

Robin Rombach, Andreas Blattmann, Dominik Lorenz, Patrick Esser, and Björn Ommer. High-resolution image synthesis with latent diffusion models. In *Proceedings of the IEEE/CVF Conference on Computer Vision and Pattern Recognition*, pp. 10684–10695, 2022.

Olaf Ronneberger, Philipp Fischer, and Thomas Brox. U-net: Convolutional networks for biomedical image segmentation. In *International Conference on Medical Image Computing and Computer-Assisted Intervention*, pp. 234–241, 2015.

Pedro Sanchez and Sotirios A. Tsaftaris. Diffusion causal models for counterfactual estimation. In *Conference on Causal Learning and Reasoning*, pp. 647–668, 2022.

Antonio Sclocchi, Alessandro Favero, and Matthieu Wyart. A phase transition in diffusion models reveals the hierarchical nature of data. *Proceedings of the National Academy of Sciences*, 122(1):e2408799121, 2025.

Jascha Sohl-Dickstein, Eric Weiss, Niru Maheswaranathan, and Surya Ganguli. Deep unsupervised learning using nonequilibrium thermodynamics. In *International Conference on Machine Learning*, pp. 2256–2265, 2015.

Jiaming Song, Chenlin Meng, and Stefano Ermon. Denoising diffusion implicit models. *arXiv preprint arXiv:2010.02502*, 2020a.

Yang Song, Jascha Sohl-Dickstein, Diederik P. Kingma, Abhishek Kumar, Stefano Ermon, and Ben Poole. Score-based generative modeling through stochastic differential equations. *arXiv preprint arXiv:2011.13456*, 2020b.

George E. Uhlenbeck and Leonard S. Ornstein. On the theory of the brownian motion. *Physical Review*, 36 (5):823, 1930.

Ashish Vaswani, Noam Shazeer, Niki Parmar, Jakob Uszkoreit, Llion Jones, Aidan N. Gomez, Łukasz Kaiser, and Illia Polosukhin. Attention is all you need. *Advances in Neural Information Processing Systems*, 30, 2017.

Kenneth G. Wilson. The renormalization group: Critical phenomena and the kondo problem. *Reviews of Modern Physics*, 47(4):773, 1975.

Jean Zinn-Justin. *Quantum Field Theory and Critical Phenomena*. Oxford University Press, 2021.

