# OpenReview forum: "Interpreting the Synchronization Gap: The Hidden Mechanism Inside Diffusion Transformers"
_TMLR — Under review for TMLR_

### Review · Reviewer_Ls6G · 2026-06-01

**Summary Of Contributions:**

This manuscript studies attention mechanism of synchronization gap in Diffusion Transformer. The authors implements Ornstein–Uhlenbeck dynamics in DiT architecture, which is analyzed through two diffusion trajectories. Experiments with DiT-XL/2 were performed, while this target should be extended to other models.

**Audience:**

Yes

**Audience Explanation:**

DiT is widely used and several researchers may find certain values in this study.

**Claims And Evidence:**

No

**Claims Explanation:**

I think the current version is not sufficient. Please see the Requested Changes below.

**Requested Changes:**

- Several theoretical findings were presented in the manuscript, including Ornstein–Uhlenbeck dynamics and symmetric Gaussian mixture, but their empirical quantity remains unmeasured. For example, local score gain gamma, routing gain, and pattern modulation gain could be measured. The theoretical findings should be supported with quantitative and qualitative empirical results.
- Experiments are based on the single model of DiT-XL/2. The observation and conclusion could be a specific property of this model, not the general DiT behavior. The target model should be extended to others. Overall, the experimental validation should be improved substantially.
- The attention decomposition, which decomposes the spatial routing term and pattern modulation term, says that the former is dominant in the low-frequency regime, but this claim is still unmeasured empirically.
- I think that the meaning of global/local and leading/trailing is inconsistent across the whole manuscript. The authors should write a clear and consistent meaning of these terms.
- From Eqs. 30-33, the cross-term is omitted as O(\epsilon^2). This requires empirical evidence: a pretrained model can have substantial residual gate.
- The bound in Eqs. 99-105 might be valid in theory but is not validated in practical DiT. The authors should measure N_{eff}^{min} and \epsilon(s) for the validity of these bounds.
- Writing should be improved.
    - global image features commits → global image features commit
    - “tab:SpecT”
    - which provide the primary mechanistic test → which provides the primary mechanistic test
    - between 16th and 22th layer → between the 16th and 22nd layers
    - these period is characterized → this period is characterized
    - Check Eq. 28
    - Protocol IIsection 3.2 → Protocol II, Section 3.2
    - An extensions to multiple classes was introduced → An extension to multiple classes was introduced
    - one of a key structural assumptions → one of the key structural assumptions
- Check references:
    - Incorrect authors for “Video generation models as world simulators”
    - Incorrect authors for “Diffpo: A causal diffusion
    model for learning distributions of potential outcomes”
    - Full author information should be presented for “Adaptive spectral feature forecasting for diffusion sampling acceleration.”

---

### Review · Reviewer_6Rav · 2026-06-05

**Summary Of Contributions:**

This paper studies a synchronization-gap phenomenon in Diffusion Transformers (DiT), following-up on or parallel to recent work by Albrychiewicz et al. on coupled diffusion processes in the context of speciation in multimodal diffusion models. The authors introduce a coupled-replica construction where two denoising trajectories are concatenated at the token level, and inter-replica attention is controlled through a coupling parameter g. This allows to study how differences between the two replicas propagate through the DiT. The paper derives an approximate model of the replica-difference attention dynamics, and looks at speciation times of the largest and smallest eigenmodes of the covariance of the replica-difference. These times are shown to differ, except for strong g -> 1 coupling. Empirically, the authors then measure speciation times, output-level coarse/fine commitment, and difference-mode energies across transformer layers, showing a similar phenomenology as in the approximate theory.

The topic is interesting, and the paper is technically ambitious. However, the current manuscript does not yet cleanly connect and explain the coupled-OU motivation, the DiT replica intervention, the local linearized attention calculation, and the empirical nonlinear DiT measurements.

### Strengths

* The coupled-replica attention construction is a nontrivial and interesting diagnostic intervention.
* Overall strong and clearly presented theory contribution.
* The very late-layer emergence of the speciation-time gap is interesting (though not explained)
* Connects statistical physics analyses of diffusion dynamics with modern transformer-based diffusion models.

### Weaknesses

* The interpretation and rationale was very unclear from the paper. It mixes several distinct levels in an for me as a reader unclear way: the synchronization gap in the prior coupled-OU for multimodal-diffusion work, the authors’ coupled-replica DiT construction looking at leading and trailing eigendirections of the difference, and empirical observations in a nonlinear pretrained DiT looking at object structure vs local details.
* Some claims appear stronger than the evidence supports. In particular, terms such as “hidden mechanism,” “intrinsic property,” and “mechanistic interpretation” suggest that the paper has identified a main operative mechanism inside DiTs, however the paper does not clearly establish the synchronization gap (as defined here) as explaining well known practical behavior.
* The theory-to-experiment link is not directly tested, e.g. the paper does not measure the terms of Eq.25 in the actual pretrained DiT (and see other requested changes).

**Additional Comments:**

/

**Audience:**

Yes

**Audience Explanation:**

Especially physicist readers working on diffusion-model theory and mechanistic interpretability of generative models will be interested in this work. The coupled-replica attention construction appears novel, and the late-layer localization of the difference-mode gap seems interesting.

**Broader Impact Concerns:**

/

**Claims And Evidence:**

No

**Claims Explanation:**

Overall, I do not doubt that the paper is technically serious and valuable. However, claims where partly difficult to interpret and the theory-experiment link was not explicit (see requested changes). After reasonable revisions I will be happy to change this rating.

**Requested Changes:**

**Major issues, most relevant for my recommendation:**

1. **Clarify the rationale and interpretation of the replica construction and synchronization gap.**
   Please explain more clearly the rationale of the coupled-replica construction, and how the measured phenomenon relates to the synchronization-gap picture in the prior coupled-OU model for multimodal diffusion. It also remained somewhat unclear to me what the practical consequences of the measured synchronization gap would be, e.g. what would be the fundamental difference if the gap did not exist? Furthermore, in the strong coupling case the gap is shown to vanish; does the strong coupling regime have a practical analogue and what is the interpretation? Analogously, what is the interpretation of the zero coupling case? I think answering these questions would strongly improve the presentation of the paper, which currently presents well the derivations, but was confusing to read on a conceptual level.

2. **Link or separate derivation, assumptions, and phenomenology more carefully.**
   Section 2.6 in seems to require assumptions about the spatial routing dominance, MLP contributions, comparable remaining model factors, and approximate mode decoupling. Please clarify whether the theory is intended as a direct approximation of the DiT dynamics (requiring to check the assumptions empirically), or as a phenomenological/toy model that can reproduce similar qualitative behavior. This distinction is also important for assessing claims such as “hidden mechanism” and “intrinsic property”.

3. **Directly test Eq. 25.**
   Since Eq. 25 is central to the interpretation, please measure the relative magnitudes of the spatial-routing term, the pattern-modulation term, and the residual of the linear approximation to the actual attention-difference update in the studied DiT. This should be reported for representative layers, timesteps, and coupling strengths. Without this, the connection between the mathematical decomposition and the empirical hidden-state dynamics remains mostly qualitative.

4. **Explain or correct the Appendix B routing-dominance argument.**
   As written, the Appendix B bound appears to contain a factor $\sqrt{N}/\sqrt{N_{\mathrm{eff}}^{\min}}$. If $N_{\mathrm{eff}}^{\min}\approx N $, this factor appears to be $O(1)$, not suppressive as $O(1/\sqrt{N_{\mathrm{eff}}^{\min}})$ by itself? Also here, if the quantities entering this bound could be measured in the actual DiT architecture, the theory-empirics link of the paper would be much stronger.

**Minor issues and optional suggestions:**

5. The claimed intermediate inversion of the global/local hierarchy appears small and noisy in the current plots. Please either provide clearer evidence and uncertainty estimates, or present this as a tentative observation of a smaller effect.

6. Since the paper argues for a general DiT-internal phenomenon, please at least include an additional ImageNet class in the appendix.

7. Maybe a layer-restricted coupling experiment could strengthen the interpretation of the late-layer effect. Since the internal gap appears mainly in the final transformer blocks, such an experiment could test whether late layers are causally responsible for the gap or are the first to reveal a difference during speciation.

8. Please make the link between output-level global/local structure and internal leading/trailing modes more explicit. Example images, difference visualizations, or a short analysis in the appendix connecting internal modes to coarse/fine output changes would make the interpretation clearer.

9. For reproducibility it would be very helpful to attach the code for the experiments, since there are many details that could potentially matter.

**Typos and small comments**

- Please recheck that the normalization convention $\sqrt{2}$ in Eq.20 is consistent afterwards; is the factor absorbed in $\delta h$ ?
- Eq.22 has a stray comma
- Eq.28: maybe a $norm$ macro is missing its backslash
- It is claimed that the residual gates remain $O(\epsilon)$ during training, how big are they on average in the pretrained DiT?
- End of p.15: "Protocol IIsection"
- Protocol I and II appear to use different sampler parameters: $\eta = 1$ and $\eta = 0$ respectively. Maybe this could be explained by a short comment.
- Caption Figs.4: "commitment scores vs coupling strength g" but x-axis is coupling duration
- p.26: "tab:SpecT" unresolved

---

### Review · Reviewer_1EDB · 2026-06-24

**Summary Of Contributions:**

The manuscript focuses on the fundamental problem of elucidating and understanding mechanisms observed in pretrained transformers. For this purpose, the authors develop an effective theoretical model, based on continuous statistical physics of coupled diffusion proceses, to understand some of these mechanisms. In particular, they focus on connecting the synchronization gap in diffusion theory to selective routing in self-attention mechanisms, to understand when this routing arises. The authors form testable hypothesis from this model, and perform numerical experiments to test these hypotheses.

**Additional Comments:**

There are no additional comments.

**Audience:**

Yes

**Audience Explanation:**

Understanding the empirical behavior of transformer architectures using well-formulated mathematical models, even when heuristic, is of interest to the community.

**Broader Impact Concerns:**

There are no broader impact concerns.

**Claims And Evidence:**

No

**Claims Explanation:**

My answer is based mostly on the organization and presentation rather than its content, and I can reconsider my assessment if they are improved. From the abstract, the findings are: "*(1) the synchronization gap is an intrinsic architectural property of DiTs that persists even when external coupling is turned off; (2) as predicted by our spatial routing bounds, the gap completely collapses under strong coupling $g \to 1$; (3) the gap is strictly depth localized, emerging sharply only within the final layers of the Transformer; and (4) global, low frequency structures consistently commit before local, high frequency details.*" I believe that the contents of the manuscript support these claims, but it is not easy to find specifically where they are reported and which arguments support them.

The document is structured in two major parts. The first develops a heuristic analysis connecting the continuous statistical physics of coupled diffusion
processes to the discrete architecture of diffusion transformers. The analysis begins in Section 2.1 with the transformer block in (13) and (14) by first proposing in (18) a modification to the attention to perform an analysis using a replica method, to then linearize around the mean and difference of two replicas, to then modify this linear model with a non-linear term to allow for speciation, to then analyze this model to explain speciation. In these steps, the document introduces heuristics that are not completely motivated, and deductions that lead to expressions that are not motivated in advance. This makes it difficult to understand which expressions are crucial for the contributions, and which ones are simply intermediate steps in the deductions. As an example, in Section 2.7 it is stated that "*latent low frequency structures commit to a basin of attraction significantly earlier than high frequency textures ($\tau_{\text{low}} < \tau_{\text{high}}$). The speciation time $\tau_{\text{spec}}$ is expected to track this transition but is not required by the theory to lie strictly between them.*" However, I cannot find the definition of these symbols, up to that point "textures" have not been a part of the analysis, and it is not clear how this specific conclusion follows from the previous deductions. Similarly, in Section 5 it is stated that "*we proved, using exact identities of the row stochastic softmax structure Appendix B, that for low frequency difference modes the spatial routing term is dominant, the pattern modulation channel exactly cancels the spatially constant component of the common mode value field, while spatial routing preserves it.*" I have difficulties understanding how the analysis in Appendix B, which has its own specific set of assumptions, leads to this claim. A simple rewrite with a few paragraphs indicating what is to be obtained at the beginning of a chain of deductions, and what are the underlying assumptions, would help the reader connect these arguments to findings (2) and (4).

The second are two experimental protocols. To my understanding, the first aims to contrast experimental results with the bifurcation predicted by the theory. The second aims to understand how the continuous synchronization gap is represented internally across the discrete Transformer depth. Once again, I believe that a simple paragraph explaining how the results explicitly relate to the previous analysis would help the reader connect them to findings (1) and (4).

**Requested Changes:**

I have some general comments.

- Please follow TMLR's editorial policies on how to reference equations.

- The manuscript presents heuristic arguments that sometimes introduce concepts that the reader may be unfamiliar with, or familiar with but in different contexts. For example, "*single mode ansatz*" in pg. 11, "*mean field model*" in pg. 10, "*discrete inverse renormalization group flow*" in pg. 12. It would help the reader if the manuscript states or references *explicitly* what these concepts are.

- Along the same lines, it would also be helpful for the reader to know what the goal, i.e., the mathematical expression or statement, of the deductions is. I suggest stating or motivating at the beginning of a chain of deductions what expression is going to be obtained, the variables it involves, and its relevance to the analysis that the authors propose.

I have also some specific comments and questions.

- In pg. 5, it is not clear how the partition is done in terms of the number of channels. Partitioning $z_t$ leads to $N$ non-overlaping blocks of dimensions $C \times p \times p$. Is it then true that $d_{\text{model}} = p^2 C$?

- In pg. 6, the point of departure for the analysis is the transformer block in (13) and (14). However, the authors then propose a non-standard normalization in (18). This modification is properly justified, but it *is* acknowledged as a departure from the standard implementation. It was not clear for this reviewer how to then interpret the analysis: if the statements are for the modified block, then how do they relate to the empirical results obtained with a standard block? This was left somewhat implicit in the text. The authors should make this explicit beyond the empirical validation of their claims.

- In pg. 6, the phrase "*these attention weights can be organized in terms of block matrices corresponding*" is incomplete.

- In pg. 7, the matrix $P_{\ell}(g)$ is defined in (26). It seems like this matrix, which involves the linearization of the softmax function in (18), would depend strongly on the point at which softmax is linearized: the differential is better conditioned when $S$ has comparable terms than when $S$ has vastly differing coefficients. This does not seem to be acknowledged in the discussion, and it is rather surprising that it seems to play no role in the analysis, e.g., in the analysis of the magnitudes of $\pi_k$. Does it play a role? If not, why is this the case?

- In pg. 8, there is an entire cross-term in (32) that is assumed negligible. The justification is that it is a product of two terms of order $\epsilon$, where $\epsilon$ appears as a size parameter controlling $\alpha_\ell$ and $\beta_\ell$. This is somewhat confusing, as the parameter for the expansion up to first order is $\|\|h_{\ell}\|\|$. What is the relation between $\|\|h_{\ell}\|\|$ and $\epsilon$? It is not immediate that second-order in $\epsilon$ is also second-order in $\|\|h_{\ell}\|\|$ and viceversa.

- In pg. 9, the local marginal distribution of the perturbation is modeled as the mixture of to Gaussians with equal covariances. This is a fair modeling choice. However, it is not clear if this is model is a good match for the experiments, nor which phenomena are poorly modeled by this choice. What would change if the covariances are not the same? What happens if there are more components?

- In Section 2.5 the authors aim to present a "*mean field approximation.*" This term has slighlty different interpretations depending on the context and, for a reader unfamiliar with the specific type of analysis being carried out, it would be useful to describe explicitly what this mean in this context.

- In Section 2.5 the authors define $\eta_k$ as the $k$-th diagonal entry of the representation of $K_g$ on the basis of eigenvectors for $V_{0, \ell}^{\top} V_{0, \ell}$. However, it is not quite clear why the off-diagonal terms play no role in the following deductions. I understand the overall goal is to reduce (41) to the system of scalar equations in (48), but the neglect of these off-diagonal terms does not seem to be justified in the text. Is this related to what the mean field approximation is in this context? If this is the case, then an explanation of when this approximation is valid is required.

- In pg. 10, it is stated that "*the empirical modes (...) must be approximately orthogonal eigenvectors of the difference covariance (...).*" This statement references an equation or symbol that does not match. For this reason, it is not clear at all why they should be approximately orthogonal.

- In pg. 10 and 11, it is stated that "*the branch separation vector (...) must be approximately aligned with the principal subspace (...).*" It is not clear at all why this *must* be the case, and the authors should discuss the consequences of this assumption, e.g., what does it mean if they are *not* aligned?

- In pg. 12, the assumption (56) is motivated as an ansatz in which "*the branch separation amplitude propagates multiplicatively.*" However, it seems that one can *always* define $G_{v,k}(s, \ell; g) = m_{k, s, \ell} / m_{k,\text{init}}$ since these are scalar quantities, and as long as $m_{k,\text{init}} \neq 0$. What are the consequences of the ansatz, e.g., what happens if it does not hold?  The manuscript then states that "*this relation is not derived from ﬁrst principles in the present work, rather, it is an empirically testable closure consistent with the linearized propagation picture.*" However, I have difficulties connecting the experimental results to this specific assumption.

- In pg. 12, the acronym RG, which I believe stands for Renormalization Group, should be defined.

- In pg. 12, it is stated that (58) is "*the discrete counterpart of the continuous SNR formula.*" Is the continuous formula (53), (54) or is it defined in a different reference?

- In pg. 13, it is stated that (59) is justified in Appendix B. However, in (59) the symbols $\chi_k$ and $\pi_k$ are not used at all, and the bounds obtained are on the ratio of the norms of $\delta AV_0$ and $A_0\delta V$. Since the definition of $\chi_k$ and $\pi_k$ are given in (47) in terms of $R$ and $P_g$, which are defined in (26), it is difficult to understand what the actual statement is.

- In pg. 13, in (53) it is stated that $\chi_{k_{\text{hi}}} > \chi_{k_{\text{lo}}}$ where $k_{\text{hi}}$ corresponds to long wavelegths, whereas $k_{\text{lo}}$ corresponds to short wavelength. However, it must be made clear what is being defined here. It could be that $k_{\text{hi}}$ and $k_{\text{lo}}$ are *defined* by (61). However, if $k_{\text{hi}}$ corresponds to indices of *long wavelengths*, which for finite samples it usually implies *low frequencies*, then implicitly $k$ is the *decreasing* ordering of eigenvalues. Note the convention in, e.g., eigenvectors of the graph Laplacian, is sometimes different.

- In pg. 13, Section 2.7 presents a summary of testable predictions. However, these predictions are stated without connecting them to specific parts of the analysis carried out up to this point. It is difficult for the reader to understand, for example, which part of the analysis supports the claim that "*since the speciation time depends on the layer the synchronization gap will not be uniform across them.*"

- In Protocol I, is the sigmoid fitted over all samples, or the medians?

- In Appendix B, the property $\delta A \mathbf{1} = 0$ seems like a consequence of the fact that $A \mathbf{1}$ is identically a constant. The first argument in the appendix could be then streamlined.

- In Appendix B, the authors motivate (102) by assuming hat "*for low frequency a constant component is dominating.*" I disagree with this statement, as it conflates *frequency* for *amplitude*. The bound in (102) only states that the average dominates the *magnitude of the fluctuations*, but imposes no constraint on their frequency. Noise-like fluctuations can have a small magnitude.

- Along the same lines, it seems that, if Appendix B assumes that ""*for low frequency a constant component is dominating*" then this has a direct impact on the statement that "*latent low frequency structures commit to a basin of attraction significantly earlier than high frequency textures*." What is the connection between these statements?